# Causal Differentiating Concepts: Interpreting LM Behavior via Causal Representation Learning

**Navita Goyal**
University of Maryland
navita@umd.edu

**Hal Daumé III**
University of Maryland
hal3@umd.edu

**Alexandre Drouin**
ServiceNow Research
Mila-Quebec AI Institute
alexandre.drouin@service.now

**Dhanya Sridhar**
Mila-Quebec AI Institute
Université de Montréal
dhanya-sridhar@mila.quebec

## Abstract

Language model activations entangle concepts that mediate their behavior, making it difficult to interpret these factors, which has implications for generalizability and robustness. We introduce an approach for disentangling these concepts without supervision. Existing methods for concept discovery often rely on external labels, contrastive prompts, or known causal structures, which limits their scalability and biases them toward predefined, easily annotatable features. In contrast, we propose a new unsupervised algorithm that identifies causal differentiating concepts—interpretable latent directions in LM activations that must be changed to elicit a different model behavior. These concepts are discovered using a constrained contrastive learning objective, guided by the insight that eliciting a target behavior requires only sparse changes to the underlying concepts. We formalize this notion and show that, under a particular assumption about the sparsity of these causal differentiating concepts, our method learns disentangled representations that align with human-interpretable factors influencing LM decisions. We empirically show the ability of our method to recover ground-truth causal factors in synthetic and semi-synthetic settings. Additionally, we illustrate the utility of our method through a case study on refusal behavior in language models. Our approach offers a scalable and interpretable lens into the internal workings of LMs, providing a principled foundation for interpreting language model behavior.

## 1   Introduction

As language models (LMs) grow more capable and complex, there is an increasing need for interpretability methods to shed light on human-interpretable factors that mediate LM behavior on a given task. Consider the following running example:

**Example 1** (Income prediction using LMs). In a set of prompts, each prompt $\mathbf{x}_n$ consists of a candidate's bio followed by an instruction asking an LM to assess whether the candidate earns a six-figure salary. Suppose $p(\text{yes}|\mathbf{x}_n)$ is *high* for bios corresponding to high-income occupations (e.g., lawyers and doctors) and *low* for bios corresponding to low-income occupations (e.g., painters and teachers). But for medium-income occupations (e.g., accountants and lecturers), the model's behavior varies by gender, leading to *higher* $p(\text{yes}|\mathbf{x}_n)$ for male-associated bios and *lower* $p(\text{yes}|\mathbf{x}_n)$ for female-associated ones.

39th Conference on Neural Information Processing Systems (NeurIPS 2025).

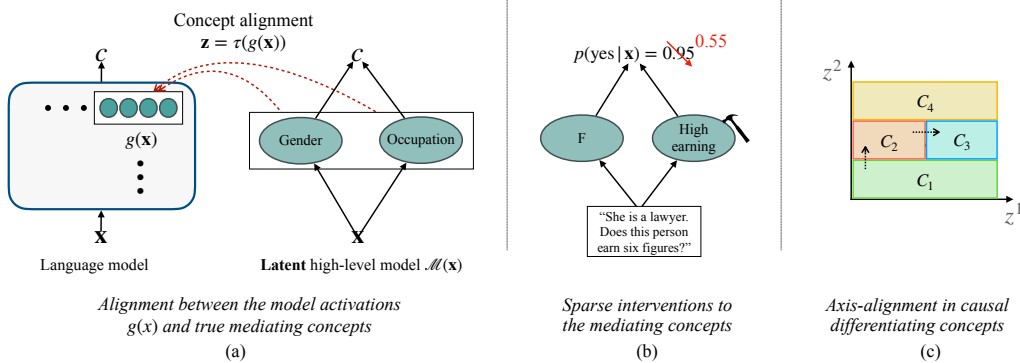

*Alignment between the model activations*     *Sparse interventions to*     *Axis-alignment in causal*
*$g(x)$ and true mediating concepts*     *the mediating concepts*     *differentiating concepts*

       (a)                             (b)                     (c)

Figure 1: $\mathcal{M}$ represents a high-level model that explains the language model's behavior $c$ in terms of latent mediating concepts $\mathbf{z}$, such that sparse interventions to mediating concepts suffice to change model behavior. Instead of assuming access to the high-level model $\mathcal{M}$, our work disentangles the learned representation $\hat{\mathbf{z}}$ with the key assumption that causal differentiating concepts are axis-aligned.

In this example, the different ranges of $p(\text{yes}|\mathbf{x}_n)$ give rise to four distinct behavior classes, and the candidate's gender and occupation are the "concepts" that mediate this behavior. These patterns may emerge due to correlations present in the data used to train the language model. This high-level model of the LM's behavior is illustrated in Figure 1(a). In this paper, we focus on a key aspect of interpretability research: finding alignments between the activations of an LM (e.g., token embeddings at different layers) and mediating concepts (e.g., gender and occupation).

The challenge in directly interpreting individual neurons or embedding dimensions as potential concepts is that LM activations generally entangle such concepts (Elhage et al., 2022; Geiger et al., 2024b). This necessitates a mapping that effectively "inverts" activations back into the space of disentangled concepts. The need for such interpretability tools has led to a suite of methods, such as linear probes (Elazar et al., 2021), contrastive activation addition (CAA) (Rimsky et al., 2024), and distributed interchange interventions (DII) (Geiger et al., 2024b) that learn mappings from activations to ground-truth concepts via various forms of supervision (e.g., concept labels, contrastive prompts, causal model over concepts). While these methods are effective at finding interpretable concepts encoded in LM activations (Wu et al., 2025), the need for supervision introduces a bias toward inferring simple concepts that we know how to annotate, such as tense, pronoun use, or language. To address these limitations and enable the discovery of behavior-relevant concepts, this paper introduces a new method for uncovering such mediating concepts from LM activations without the need for supervision from ground-truth concept labels.

Since fully unsupervised learning is not identifiable (Hyvärinen and Pajunen, 1999; Locatello et al., 2019)—i.e., there are infinitely many correct solutions—many practical approaches to unsupervised learning introduce inductive biases. Sparse autoencoders (Huben et al., 2024, SAEs) seek to recover *all* human-interpretable features encoded in LM activations by constraining the feature representation to be sparse. However, SAEs require post hoc analysis both to interpret individual features (for instance, by feeding a set of examples that activate a feature into a large language model to infer its meaning) and to identify which features affect model behavior (for instance, by manipulating different features, one at a time, and observing the corresponding effects on model behavior) (Bills et al., 2023; Bricken et al., 2023; Paulo et al., 2024). Moreover, since SAEs seek to invert potentially billions of concepts from activations, they may not be able to uniquely recover many concepts (Menon et al., 2025). As one example, SAEs could decompose a concept (e.g., "*Einstein*") into a combination of features (such as "*scientist*", "*Germany*", and "*famous person*") (Leask et al., 2025). Even though in this case the decomposition is interpretable, it may make concepts harder to intervene on for causal insights. This makes SAEs cumbersome to use for understanding targeted model behavior. In this work, we place the interpretation of model behavior front and center, aiming to learn concepts at the level of granularity that is directly relevant to codify the behavior under consideration.

Concretely, we learn mappings from model activations $g(\mathbf{x})$ to features $\hat{\mathbf{z}} = \tau(g(\mathbf{x}))$, so that the learned features $\hat{\mathbf{z}}$ align with the true mediating concepts (e.g., gender and occupation in Example 1). To arrive at an identifiable objective, we first introduce the idea of "causal differentiating concepts."

Put simply for now, causal differentiating concepts are the concepts whose values we *must* change for any example to elicit a different model response. In Example 1, we *must* change the candidate's occupation if we want the model to change $p(\text{yes}|\mathbf{x}_n)$ from a high to a low value. Motivated by work on identifiable representation learning that leverages sparse effects of features on outcomes of interest (Lachapelle et al., 2023), we make the key assumption that causal differentiating factors are sparse—i.e., sparse changes suffice to change behavior. We encode this assumption into a constrained contrastive learning objective that we prove recovers disentangled concepts that mediate some model behaviors of interest.

To summarize our contributions: (1) We formalize concept discovery in settings where high-level mediating factors are unknown. We introduce causal differentiating concepts—factors that must change to elicit a different model behavior—and propose a sparsity assumption that enables their identification. (2) We develop a constrained contrastive learning objective that enforces this assumption and can provably recover disentangled, interpretable features. (3) We validate our method in both controlled experiments and a real-world case study, where the underlying causal factors are unknown and the assumption is unverifiable, demonstrating the potential of our approach in practice.

## 2   Problem setting

We consider the setting where a language model takes in an input sequence $\mathbf{x} \in \mathcal{X}$ and outputs a sequence $\mathbf{y} \in \mathcal{Y}$. The fine-grained responses $\mathbf{y}$ are categorized into $m$ coarse-grained *behavior* classes $\{1, \ldots, m\}$ so that each input $\mathbf{x}$ is associated with a discrete behavior label $c$. For instance, when studying refusal behavior in language models, all queries $\mathbf{x}$ to the model that elicit responses such as "*I am sorry...*", or "*I cannot respond...*" are mapped to the same *refusal* behavior class. These categorizations of fine-grained responses $\mathbf{y}$ into coarse-grained behaviors can be provided entirely by a domain expert or by clustering the next-token probabilities $p(\mathbf{y}|\mathbf{x})$ learned by the model, as with causal feature learning (Chalupka et al., 2017).

Motivated by work on abstracting neural networks (Geiger et al., 2021, 2024a), we assume a high-level model $c = \mathcal{M}(\mathbf{x})$ that explains the LM's behavior $c$ in terms of $k$ discrete mediating concepts $\mathbf{z}$, so that $c \perp\!\!\!\perp \mathbf{x}|\mathbf{z}$. Figure 1 illustrates a high-level model for Example 1, where the model's likelihood of predicting *yes* is mediated by two latent concepts $\mathbf{z}$: the gender and occupation of a candidate.

**Problem.**  To interpret model behavior, the goal is to map model activations $g(\mathbf{x})$ to $k$ features $\hat{\mathbf{z}} = \tau(g(\mathbf{x}))$, via a learned encoder $\tau$, so that the learned features $\hat{\mathbf{z}}$ align with the true mediating concepts $\mathbf{z}$ (e.g., gender and occupation).

Given input-behavior pairs $(\mathbf{x}, c)$, it would be tempting to simply find the most activated neurons or token embedding dimensions among examples in a given class and use these as proxies for the mediating concepts $\mathbf{z}$. However, such activations typically entangle interpretable concepts like gender or occupation (Elhage et al., 2022; Geiger et al., 2024b). To find an alignment between activations and concepts, Geiger et al. (2024b) propose using a fully known high-level model $\mathcal{M}(\mathbf{x})$ to supervise the learning of $\tau(g(\mathbf{x}))$.

**Key idea.**  To overcome the need to fully specify $\mathcal{M}(\mathbf{x})$, this paper proposes weaker assumptions that we can make about high-level concepts and behaviors that drive identifiable concept learning.

## 3   Learning causal differentiating concepts

Our method operationalizes two assumptions that we make about the high-level model $c = \mathcal{M}(\mathbf{x})$. The key assumption, as illustrated in Figure 1(c), says, loosely speaking, that we can change a model's behavior with only a sparse change to the mediating concepts $\mathbf{z}$. We formalize both assumptions and the resulting learning objective for recovering mediating features $\hat{\mathbf{z}} = \tau(g(\mathbf{x}))$ that align with the true underlying mediating concepts $\mathbf{z}$. Crucially, we show that because of the assumptions, the mapping $\tau(g(\mathbf{x}))$ becomes identifiable, meaning that the recovered $\hat{\mathbf{z}}$ features are guaranteed to correspond to the mediating concepts $\mathbf{z}$ up to permutation and scaling indeterminacies.

> **Assumption 1.**  The true conditional probability of a behavior given the mediating concepts $\mathbf{z}$ encoded by the high-level model $c = \mathcal{M}(\mathbf{x})$ is $p(C = c|\mathbf{z}) \propto \exp(\mathbf{w}_c^\top \mathbf{z})$. That is, the model's behavior $c$ for an input $\mathbf{x}$ is related to the mediating latent variables $\mathbf{z}$ by a logit-linear function.

This assumption, used in other works on disentangled representation learning (Ahuja et al., 2022b), can be motivated by the structure of language models, where the final layer embedding linearly influences next-token logit probabilities. Here, we extend such a logit-linear assumption to behaviors.

To formalize the assumption that sparse feature changes enable behavior changes, we define the notion of interchange intervention and causal differentiating concepts. First, we define $\mathbf{z}^S$ as a subset of the vector $\mathbf{z}$ that selects elements of the set $S$, $C_k = \{\mathbf{x}|c = k\}$ as the set that contains inputs with behavior label $k$, and $\mathbf{x}_k \in C_k$ as an input in this set.

**Definition 1.** For two inputs $\mathbf{x}_k \in C_k$ and $\mathbf{x}_l \in C_l$ such that $k \neq l$, an **interchange intervention** on a subset of mediating concepts $S \subset \{1, \ldots, d\}$ in the high-level model is defined as $\mathcal{M}_{\mathbf{z}_k^S \leftarrow \mathbf{z}_l^S}(\mathbf{x}_k)$ where for $r \in S$, the $r$-th component of the high-level representation $\mathbf{z}_k$ (associated with the input $\mathbf{x}_k$) is replaced with the corresponding value from $\mathbf{z}_l$ from $\mathbf{x}_l$.[1]

**Definition 2.** A set of latent mediating factors $\mathbf{z}^S$ is defined as **causal differentiating concepts** between two groups $C_k$ and $C_l$ if, for some inputs $\mathbf{x}$ that are labeled with class $k$, an interchange intervention on $S$ is both necessary and sufficient to change the behavioral label from $k$ to $l$.

(*Necessary condition*)    For any $\mathbf{x}_k \in C_k$ and $\mathbf{x}_l \in C_l$, $\mathcal{M}_{\mathbf{z}_k^{S'} \leftarrow \mathbf{z}_l^{S'}}(\mathbf{x}_k) \neq c_l$,    where $\bar{S}' \cap S \neq \varnothing$ 

$$(1)$$

(*Sufficient condition*)    For some $\mathbf{x}_k \in C_k$ and $\mathbf{x}_l \in C_l$, $\mathcal{M}_{\mathbf{z}_k^S \leftarrow \mathbf{z}_l^S}(\mathbf{x}_k) = c_l.$    $$(2)$$

A latent mediating factor $r$ is thus the **1-sparse causal differentiating concept** between a pair of group $C_k$ and $C_l$ if the set of causal mediating concepts between $C_k$ and $C_l$ is $S = \{r\}$.

> **Assumption 2.** Every dimension $z^r$ is a 1-sparse causal differentiating concept for some pair of groups $C_k$ and $C_l$.

**Corollary 1.** For the groups $C_k$ and $C_l$, consider a 1-sparse latent mask $\boldsymbol{\delta}_{kl} \in \mathbb{R}^d$, which is a $d$-dimensional vector with a nonzero value at position $r$ corresponding to the causal differentiating concept between the two groups and zeros elsewhere. It follows from the necessary and the sufficient conditions of causal differentiating concepts that for some $\mathbf{x}_k \in C_k$ and $\mathbf{x}_l \in C_l$, $\mathbf{z}_l = \mathbf{z}_k + \boldsymbol{\delta}_{kl}$.[2]

**Method.**    To learn the mapping $\hat{\mathbf{z}} = \tau(g(\mathbf{x}))$ from activations $g(\mathbf{x})$ to interpretable concepts, we introduce a constrained contrastive learning objective designed to satisfy the key assumptions (Assumptions 1 and 2) about the true mediating concepts $\mathbf{z}$ in the high-level model $\mathcal{M}(\mathbf{x})$. We implement $\hat{\mathbf{z}} = \tau(g(\mathbf{x}))$ as a bottleneck layer on top of the language model's final layer, ensuring that the dimension of $\hat{\mathbf{z}}$ is less than the dimension of $g(\mathbf{x})$.

Our objective has two components. The first ensures that this bottleneck representation extracts information that is predictive of the labeled behavior $c$, using a log-linear predictor $h(\tau(g(\mathbf{x})))$ to enforce Assumption 1. The bottleneck serves to filter out irrelevant factors, retaining only information necessary for predicting $c$. Specifically, we minimize the categorical cross entropy loss:

$$\min_{\tau,h} \mathbb{E}_{p(\mathbf{x},y)} \big[ -\log h(\tau(g(\mathbf{x})))_y \big]. \tag{3}$$

The second contrastive loss term satisfies Assumption 2 by using Corollary 1. Essentially, for each pair of groups $C_k$ and $C_l$, the learner guesses a 1-sparse perturbation $\hat{\boldsymbol{\delta}}_{kl}$ and searches for a pair of examples $\mathbf{x}_i \in C_k$ and $\mathbf{x}_j \in C_l$ such that $\hat{\mathbf{z}}_j = \hat{\mathbf{z}}_i + \hat{\boldsymbol{\delta}}_{kl}$. Formally, the objective is,

$$\min_{\tau,\delta} \sum_{k,l} \Big[ \min_{\substack{\mathbf{x}_i \in C_k \\ \mathbf{x}_j \in C_l}} \mathbb{E}[||\tau(g(\mathbf{x}_j)) - \tau(g(\mathbf{x}_i)) - \hat{\boldsymbol{\delta}}_{kl}||^2] \Big]. \tag{4}$$

Since Assumption 2 only requires that each concept $z^r$ is a 1-sparse causal differentiating concept for at least one pair of groups, we do not expect to find a 1-sparse perturbation $\hat{\boldsymbol{\delta}}_{kl}$ between all pairs

---

[1]Notation: We use a subscript for enumerating the sample and a superscript for enumerating the dimension. So $\mathbf{z}_i$ is the true latent for the $i^{th}$ sample and $z_i^r$ is the $r^{th}$ dimension of this latent variable.

[2]This is true because if $\boldsymbol{\delta}_{kl}$ were nonzero in more than one position $\forall \mathbf{x}_k \in C_k$ and $\mathbf{x}_l \in C_l$, the sufficiency condition would be violated. Conversely, if $\boldsymbol{\delta}_{kl}$ were zero everywhere, the necessary condition would be violated.

of groups. However, as we show later, a disentangled solution $\tau(\hat{g}(\mathbf{x}))$ achieves a lower objective value than an entangled solution, which generally captures fewer axis-aligned changes across pairs of groups. In practice, we search over possible $\delta_{kl}$ and perform the inner minimization over $i$ and $j$ by sampling multiple $\mathbf{x}_i \in C_k$ and $\mathbf{x}_j \in C_l$. To ensure that the learned perturbations cover all mediating concepts, we constrain $\mathrm{span}_{k,l}(\hat{\boldsymbol{\delta}}_{kl}) = d$.

We show that under the above assumptions, our method identifiably recovers interpretable causal factors up to permutation and scaling.

**Theorem 1.** *If the Assumption 1 holds, then the function $\hat{\tau}$ that satisfies Equation* (3) *gives us the true latents up to an affine transformation.*

**Theorem 2.** *If Assumptions 1 and 2 hold, then the function $\hat{\tau}$ that satisfies Equations* (3) *and* (4) *identifies the true latents up to permutation and scaling.*

Intuitively, we get the result in Theorem 1 because of the log-linearity in the prediction function (Ahuja et al., 2022b). For Theorem 2, we leverage the fact that for all causal differentiating concepts of interest, by Corollary 1, there exist some pairs $\mathbf{x}_i$ and $\mathbf{x}_j$ that are related by a sparse latent shift, allowing us to adapt the proof from Ahuja et al. (2022a). See the full proofs in Appendix A.

# 4  Experimental details

We evaluate the ability of our method to disentangle mediating concepts in settings where we have some domain knowledge about what the desired mediating concepts are. We compare our method to baselines without disentanglement guarantees and sparse autoencoders (SAEs), and find that our method outperforms these related methods.

**Data.**  We conduct our experiments in three settings: (1) synthetic data, (2) semi-synthetic data with real text and synthetic labels, and (3) non-synthetic data with text and LM outputs.

Synthetic and semi-synthetic datasets allow us to control the ground-truth causal factors and their influence on outcomes, enabling a precise evaluation of the ability of our proposed method to recover the true causal factors up to permutation and scaling. The semi-synthetic data with text inputs enables testing our method in the context of language models, assessing whether our method can isolate causally relevant factors from the many encoded during LM pretraining. However, it is limited to naturally occurring features in the text. Fully synthetic data offers more control: we can vary complexity, sparsity, and the number of causal factors. Lastly, we present a case study using a dataset with queries with different harmfulness categories and study the language model's refusal behavior, demonstrating how our method can be implemented and evaluated in practical, in-the-wild scenarios.

***Synthetic data.***  For synthetic data, we consider true factors $\mathbf{z} \sim \mathcal{N}(\mu, \sigma) \subset \mathbb{R}^d$ with $d = 2$ and $d = 3$. We relate factors $\mathbf{z}$ to behavior labels $c$, such that the resulting data satisfy Assumption 2. The resulting behavior groups are illustrated in Table 1, with each color representing a different group.

We generate $\mathbf{x} \in \mathbb{R}^n$ given the factor $\mathbf{z}$ using linear and non-linear mixing functions. Moran et al. (2022) show identifiability up to permutation and scaling for non-linear sparse mixing functions, where each component $\mathbf{x}^j$ depends only on a subset of factors. We also experiment with non-linear non-sparse mixing functions to assess whether our method yields identifiability when mapping from factors to observations is more complex. The list of mixing functions is included in Appendix B.

***Semi-synthetic data.***  For semi-synthetic data, we consider the data generating process described in Example 1. We use the bios from the BiasBios dataset (De-Arteaga et al., 2019) as the textual input $\mathbf{x}$ and generate an outcome $\mathbf{y}$ that represents, e.g., a model responding "yes" or "no" to whether or not the candidate makes six figures. We consider two causal factors—binary gender (male/female) and occupation level (high/medium/low). We restrict our experiments to binary gender because the BiasBios dataset provides binary labels, which we use as the gender factor. For occupation, we categorize the occupation associated with each bio into three categories—high income (e.g., doctors, lawyers), medium income (e.g., nurses, accountants), and low income (e.g., paralegal, painters).[3] This grouping allows us to test whether our method can recover causal factors that influence the outcome $\mathbf{y}$ at the appropriate level of abstraction, rather than solely relying on the semantic cues from the bios $\mathbf{x}$.

---

[3]These groupings are derived from the U.S. Department of Labor's Employment and Earnings by Occupation statistics (https://www.dol.gov/agencies/wb/data/occupations).

We simulate model behavior $c$ such that $c$ is mediated entirely by the true causal factors $\mathbf{z}$. We design $p(\mathbf{y}|\mathbf{x})$ so that we can derive labels for high-level model behavior $c$ by simply clustering $p(\mathbf{y}|\mathbf{x})$. The resulting groups are the same as in the synthetic data experiments with $d = 2$ (Table 1; *top*). Thus, we get $z^1 =$ gender and $z^2 =$ occupation with groups $\{\{O\text{=High}\}, \{O\text{=Med}, G\text{=Male}\}, \{O\text{=Med}, G\text{=Female}\}, \{O\text{=Low}\}\}$.

***Non-synthetic data.*** For our case study experiments, we consider refusal behavior in models. We use a collection of harmful prompts, sampled from MALICIOUSINSTRUCT (Huang et al., 2024), HARMBENCH (Mazeika et al., 2024), ADVBENCH (Zou et al., 2023), and TDC2023 (Mazeika et al., 2022), harmless prompts, sampled from ALPACA (Taori et al., 2023), and pseudo-harmful prompts, sampled from OR-BENCH (Cui et al., 2025). We use common refusal patterns in Llama-3.1-8B model, such as "*I am sorry*" and "*I cannot*" to extract $p(\text{refusal}|\mathbf{x})$. We cluster model behavior into three classes based on $p(\text{refusal}|\mathbf{x})$, which serve as the behavior classes that we aim to explain. We train $\tau$ to obtain the *aligned* hidden representations $\mathbf{z} \in \mathbb{R}^2$. Note that the model has no access to the actual class labels of harmful, pseudo-harmful, and harmless prompts. We visualize the learned representation $\mathbf{z}$ for these three prompt sets to understand what factors affect model refusal behavior.

**Models.** We conduct the semi-synthetic and non-synthetic experiments with three language models—DistilBert-base (66M) (Sanh et al., 2020), Llama-3.1-8B (Touvron et al., 2023) and Qwen2-7B (Bai et al., 2023). For synthetic data, where the input $\mathbf{x}$ is not text, we replace the language model encoder with a feedforward neural network. To obtain $g(\mathbf{x})$, we fit a variational autoencoder to $\mathbf{x}$.

We compare our method against two baselines that use autoencoding and prediction objectives without the contrastive constraint. The autoencoding baseline trains the bottleneck $\tau$ to reconstruct $\mathbf{x}$ to mimic vanilla contrastive learning without constraints, and the prediction objective encourages discarding irrelevant information but, crucially, both baselines are not guaranteed to identify the true mediating concepts. Additionally, we perform detailed comparisons of our method with sparse autoencoder baselines, discussed in Section 6. More implementation details are included in in Appendix B.

**Evaluation metrics.** We evaluate the effectiveness of our method at recovering the ground-truth causal factors using the disentanglement-completeness-informativeness (DCI) metrics (Eastwood and Williams, 2018). Briefly, disentanglement measures the extent to which a representation disentangles the underlying factors of variation, that is, whether each feature capture at most one causal factor. Completeness measures the extent to which each causal factor is captured by a single learned feature. Informativeness measures the amount of information that a representation as a whole captures about the underlying factors of variation. Thus, a rotated but disentangled representation will have an informativeness score of $1.0$. We also include the Mean Correlation Coefficient (MCC) metric (Khemakhem et al., 2020), which computes the maximum linear correlations (accounting for permutations in the learned representations), giving a measure of disentanglement.

## 5 Results

Table 1 shows a comparison between our method and the baseline methods for synthetic data. We see that, across all data-generating functions, all methods achieve near-perfect informativeness scores. However, as expected, the baseline methods entangle the true causal factors, leading to low disentanglement and completeness scores. In contrast, our method achieves significantly better disentanglement, with DCI-D scores exceeding $0.89$ and MCC scores above $0.86$ across all settings.

Next, Table 2 shows results on the semi-synthetic data. We find that in the semi-synthetic setting, the autoencoding baseline shows a low informativeness scores. This is expected since textual data contains a large amount of information, and without additional signals, the autoencoding baseline may struggle to determine which information to retain, potentially discarding information relevant to the true causal factors. In contrast, adding the prediction objective results in a boost in the informativeness score across all models. However, both baselines exhibit low disentanglement and completeness scores. In comparison, incorporating our contrastive constraint consistently improves the disentanglement metrics (MCC and DCI-D scores) across all evaluated models.

Lastly, for our case-study experiments, since we do not have ground-truth causal factors, we instead visualize the learned latent space for the different sets of prompts. Figure 2 shows the learned latent space $\hat{\mathbf{z}} = \tau(g(\mathbf{x}))$ for Llama-3.1-8B (the findings for Qwen2-7B are consistent and included in Appendix C). Consistent with our previous analysis, we find that the autoencoding baseline is not

Table 1: Synthetic data. *(Left):* Example of data with $d = 2$ (top) and $d = 3$ (bottom) latent factors. *(Right):* Disentanglement (D), Completeness (C), Informativeness (I), and Mean Correlation Coefficient (MCC) scores with latent dimension $d = 2$ (top; shaded) and $d = 3$ (bottom; unshaded).[4]

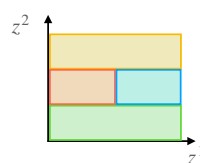

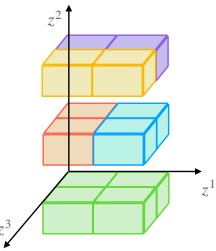

| Mixing fn | Method | MCC | D | C | I |
|---|---|---|---|---|---|
| Linear | Autoencoding | 0.76 | 0.30 | 0.36 | 1.0 |
| | Autoenc+Pred | 0.60 | 0.21 | 0.21 | 1.0 |
| | Our constraints | **0.99** | **1.0** | **1.0** | 1.0 |
| Non-linear, sparse | Autoencoding | 0.81 | 0.24 | 0.33 | 1.0 |
| | Autoenc+Pred | 0.72 | 0.0 | 0.0 | 1.0 |
| | Our constraints | **0.91** | **0.97** | **0.97** | 1.0 |
| Non-linear, non-sparse | Autoencoding | 0.67 | 0.0 | 0.0 | 1.0 |
| | Autoenc+Pred | 0.76 | 0.02 | 0.02 | 1.0 |
| | Our constraints | **0.92** | **0.90** | **0.92** | 1.0 |
| Linear | Autoencoding | 0.77 | 0.34 | 0.35 | 1.0 |
| | Autoenc+Pred | 0.65 | 0.23 | 0.31 | 0.87 |
| | Our constraints | **0.94** | **0.99** | **0.99** | 1.0 |
| Non-linear, sparse | Autoencoding | 0.74 | 0.19 | 0.25 | 1.0 |
| | Autoenc+Pred | 0.77 | 0.54 | 0.51 | 1.0 |
| | Our constraints | **0.90** | **0.92** | **0.95** | 1.0 |
| Non-linear, non-sparse | Autoencoding | 0.78 | 0.51 | 0.60 | 1.0 |
| | Autoenc+Pred | 0.86 | 0.54 | 0.59 | 1.0 |
| | Our constraints | **0.89** | **0.89** | **0.89** | 1.0 |

Table 2: Disentanglement (D), Completeness (C), Informativeness (I), and Mean Correlation Coefficient (MCC) scores on semi-synthetic data.[4]

| Model | Method | MCC | D | C | I |
|---|---|---|---|---|---|
| Distilbert-base | Autoencoding | 0.17 | 0.01 | 0.01 | 0.57 |
| | Autoenc+Pred | 0.32 | 0.18 | 0.20 | 0.85 |
| | Our constraints | **0.85** | **0.91** | **0.92** | **0.99** |
| Qwen2-7B | Autoencoding | 0.32 | 0.01 | 0.04 | 0.57 |
| | Autoenc+Pred | 0.69 | 0.06 | 0.06 | 0.83 |
| | Our constraints | **0.79** | **0.80** | **0.84** | **0.97** |
| Llama-3.1-8B | Autoencoding | 0.06 | 0.0 | 0.0 | 0.49 |
| | Autoenc+Pred | 0.54 | 0.05 | 0.11 | 0.84 |
| | Our constraints | **0.81** | **0.86** | **0.88** | **0.98** |
| | Sparse-autoencoders | 0.51 | 0.49 | 0.52 | 0.56 |

able to distinguish between different data distributions. Adding the prediction constraint for the class labels based on $p(\mathbf{y} = \text{refusal}|\mathbf{x})$ leads to distinctive clustering of the harmful, pseudo-harmful, and harmless prompts. However, the learned representation is not axis-aligned. For instance, the representation for harmful and pseudo-harmful prompts differ along both latent dimensions, similarly for pseudo-harmful and harmless prompts. In contrast, our method yields an axis-aligned latent space. These directions can be interpreted as *harmfulness* (dimension 1) and *topic* (dimension 2).

## 6 Comparison to sparse autoencoders

We compare our approach to sparse autoencoders in the semi-synthetic setting. We use LlamaScope (He et al., 2024), a popular resource with 256 SAEs trained on each layer and sublayer of the Llama-3.1-8B model, with 32k and 128k features. We restrict our experiments to the residual stream SAEs as they are reported to perform best across all metrics evaluated in the original paper, resulting in 64 SAEs (one for each 32k and 128k feature dimension and all 32 layers of Llama-3.1-8B).

**Evaluation.** We perform 3 evaluations for both the 32k- and 128k-feature SAEs. First, we measure disentanglement and informativeness for the full feature vectors, which reflect overall sparsity and predictiveness of SAE features. Since the feature dimensionality of SAEs far exceeds the number of

---

[4]The numbers in bold indicate statistical significance at $p < 0.05$. Details on the analysis in Appendix B.

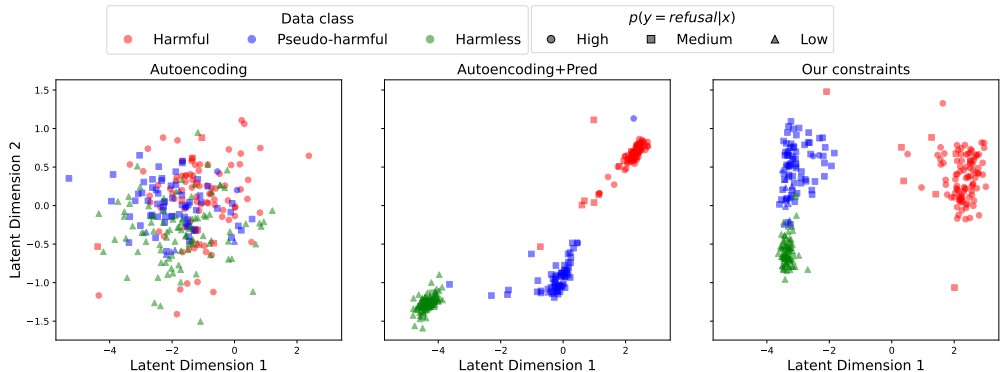

Figure 2: Latent space for refusal behavior in Llama-3.1-8B with autoencoding (*left*), autoencoding + prediction (*center*), and our constraints (*right*). The baseline method entangles the two latent dimensions, but adding the contrastive constraint leads to an aligned latent space.

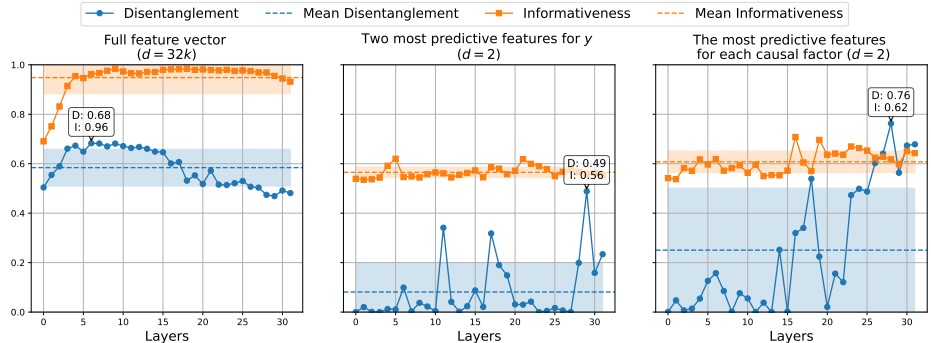

Figure 3: Disentanglement and informativeness scores for 32k-dimensional SAEs across model layers. Solid lines indicate layer-wise scores, dashed lines denote the mean, and shaded regions represent $\pm 1$ standard deviation. Each subplot is annotated with the maximum disentanglement score along with its corresponding informativeness score.

ground-truth latent factors, we also train a linear classifier on the SAE features to predict $c$. From this, we select the two most predictive features and compute disentanglement and informativeness scores on this reduced 2-dimensional representation, matching the number of ground-truth latents required to predict $c$. This evaluation most closely aligns with our setting, which is designed to learn features that best predict $c$. Finally, we perform a ceiling evaluation by training two separate classifiers—one for each ground-truth factor—and identifying the most predictive feature for each. We then compute disentanglement and informativeness scores on the resulting 2-dimensional feature vector.

**Results.** The results for 32k-dimensional SAEs are shown in Figure 3. SAE features exhibit high informativeness across multiple layers, with a mean and standard deviation of $0.95 \pm 0.07$. However, disentanglement scores remain relatively low ($0.58 \pm 0.07$), with a maximum of $0.68$.

When considering only the two most informative features for predicting $c$, both informativeness ($0.57 \pm 0.02$) and disentanglement scores ($0.08 \pm 0.12$) drop significantly. This suggests that SAEs do not isolate the true causal factors into two distinct features. Given the low informativeness in the top-2 features analysis, we hypothesize that SAE features may be more fine-grained than the true causal factors (Leask et al., 2025). To investigate this, we evaluate disentanglement by selecting the top-$k$ most informative features for predicting $c$ such that the overall informativeness is at least $0.95$. Even under this setting, disentanglement scores remain low ($0.41 \pm 0.04$) with a maximum of only $0.46$. Finally, the rightmost plot in Figure 3 shows a ceiling analysis using the most predictive features for each of the two ground-truth causal factors. Even under this best-case feature selection, the maximum disentanglement score observed is $0.76$, which remains notably lower than the score achieved by our method ($0.86$) on the Llama-3.1-8B model. The results are consistent for the 128k-dimensional SAE, as shown in Figure 8 in the appendix.

Beyond the empirical comparisons, we note some key differences between our method and sparse autoencoders. Notably, SAEs do not offer a natural mechanism for identifying which of the 32k or 128k features, across 4 activations and 1024 tokens, are relevant to a specific model behavior. While we explore several strategies for selecting the most relevant SAE features, the process remains non-trivial and ad hoc. Furthermore, a direct comparison is limited, as SAEs are not explicitly trained to predict the behavior class $c$. However, in certain settings, such behavior-targeted interpretation may be more valuable, which our work is designed for.

## 7    Sensitivity to assumptions and implementational choices

**Choice of latent dimensionality.**    In the experiments discussed in Section 5, we fix the number of the learned latent dimensions to that of true mediating concepts. However, the number of true mediating concepts is unlikely to be known beforehand. To account for this, we perform additional experiments by varying the dimensions of learned latents to assess how over-specification or under-specification affects disentanglement metrics and whether these could be used as a signal to adjust the specified latent dimensionality. The details of experiments and results are included in Appendix C. We find that over-specifying latent dimensionality consistently leads to a drop in completeness scores and under-specifying latent dimensionality consistently leads to a lower disentanglement score. These measures can thus serve as useful signals for adjusting the latent dimensionality as a hyperparameter.

We restrict this sensitivity analysis to synthetic experiments because a parallel comparison is not feasible in semi-synthetic and non-synthetic cases as the dimensionality of true mediating concepts is dataset-dependent and cannot be dictated externally. We believe that the insights from synthetic experiments can be extended to a general setting to determine latent dimensionality. Even though non-synthetic settings lack ground-truth mediating concepts, not allowing direct evaluation of disentanglement and completeness scores, the disentanglement measures can be estimated across multiple model runs as a signal for guiding the latent dimensionality, following Duan et al. (2020).

**Choice of intervention layer.**    In the experiments discussed in Sections 5 and 6, we apply our constrained learning objective to the representations at the final layer of the LLM, where the linearity assumption is most naturally satisfied. However, our method can also be applied to intermediate layers. The proposed contrastive learning constraints (Equation (4)) are designed to disentangle linearly mixed concepts, an assumption that may not strictly hold for intermediate representations. Nevertheless, recent studies support the *linear representation hypothesis*, suggesting that latent concepts are approximately linearly encoded in language model representations (Roeder et al., 2021; Jiang et al., 2024; Park et al., 2024). Based on this, we hypothesize that our method can be used for unsupervised discovery of latent concepts in the model's internal layers. To empirically test this, we extend our refusal case study to intermediate representations and examine whether our method can extract relevant insights from those layers. Detailed results are included in Appendix C. Overall, we find that our approach does not yield disentangled representations in the early layers (up to layer 9 in the Llama-3.1-8B model), but past layer 9, the structure presented in Figure 2 starts to emerge. That is, we observe that the harmfulness and topicality dimensions identified in our experiments can also be disentangled in intermediate model representations.

**Violation of 1-sparsity assumption.**    Our main theoretical and experimental results assume 1-sparsity in causal differentiating concepts. We note that our 1-sparse assumption is more relaxed than that of Ahuja et al. (2022a) since we do not require that every pair of clusters has only 1 causal differentiating concept or that the pair of clusters that have 1 causal differentiating concepts are known beforehand, rather that every $z^r$ is 1-sparse for some pair of clusters. We expect our results can be further extended to a 2-sparse condition (or generally speaking, a p-sparse condition) following the proof strategy as in Ahuja et al. (2022a). We include some preliminary experiments to test this in Appendix C, indicating that our method can be extended beyond 1-sparse settings as well. However, we leave formal extensions of our approach to p-sparse causal differentiating concepts to future work.

## 8    Related work

**Causal representation learning.**    This work presents an identifiable approach to learning concepts from observed LM activations, extending ideas from the field of causal representation learning (CRL)—

see Yao et al. (2025) for a comprehensive overview. In brief, CRL methods enjoy identifiability guarantees by leveraging paired datasets (Zhang et al., 2023; Ahuja et al., 2024) or samples (see below), auxiliary labels (Roeder et al., 2021; Khemakhem et al., 2020; Rajendran et al., 2024), or extra assumptions about the data-generating process, such as sparse decoding (Moran et al., 2022; Gresele et al., 2021). In this paper, we take inspiration from a line of CRL works that leverage sparsity assumptions such as sparse transitions in latent temporal models (Lachapelle et al., 2022, 2024), sparse latent perturbations across samples (Ahuja et al., 2022a; Brehmer et al., 2022; Locatello et al., 2020; Joshi et al., 2025), or sparse dependencies between labels and features (Lachapelle et al., 2023). Here, we introduce a new assumption on sparse causal differentiating factors, in effect finding "pseudo" counterfactual pairs of samples $\mathbf{x}$ that vary sparsely in concepts.

**Causal abstraction.** Causal abstraction (also known as causal feature learning) aims to abstract low-level features (microvariables) into high-level features (macrovariables) such that the causal effect of intervention in the low-level model corresponds to the causal effect of corresponding interventions in the high-level model (Chalupka et al., 2017; Rubenstein et al., 2017; Beckers and Halpern, 2019; Beckers et al., 2020). Geiger et al. (2020, 2021) adapt causal abstraction for mechanistic interpretability of neural networks by aligning neurons to high-level features in a human-interpretable hypothesized algorithm. Since concepts are typically distributed across neurons, Geiger et al. (2024b) propose learning alignments between concepts and LM activations, using a known high-level causal model to supervise the learning, following Geiger et al. (2022); Wu et al. (2023). We take inspiration from causal abstraction in this work to align concepts and LM activations, but require weaker assumptions than assuming that the causal model is known.

**Concept discovery and influence.** There are a host of methods in machine learning, and language modeling in particular, for interpreting learned concepts and assessing their influence on model behavior. Broadly, these methods fall into two categories—(1) *supervised methods*, which identify predefined concepts within model activation space, for instance, by aligning model activations with concepts using examples labeled with predefined concepts (Koh et al., 2020), or identifying concept directions in activation space (Kim et al., 2018; Elazar et al., 2021; Ravfogel et al., 2022a,b; Belrose et al., 2023) using concept-conditional examples, and (2) *unsupervised methods*, which discover latent structure in model activations, for instance, by clustering contextual representations (Dalvi et al., 2022; Sajjad et al., 2022), or finding examples that highly activate a feature (and those that do not) and feeding them into an LM to label the feature (Bills et al., 2023; Kalibhat et al., 2023).

Due to polysemanticity in language models (Elhage et al., 2022), however, individual neurons often lack consistent semantic interpretation. To address this, sparse autoencoders disentangle features by learning a sparse intermediate representation (Huben et al., 2024; Bricken et al., 2023), which can again be interpreted by finding examples that highly activate a feature and feeding them into an LLM for labeling. The effect of these features on model behavior is studied using activation patching (Huben et al., 2024), feature clamping (Bricken et al., 2023), logit weight inspection (Bricken et al., 2023), training linear probes (Rao et al., 2024), vocabulary projection (Gur-Arieh et al., 2025). Unlike our approach, these methods offer post-hoc, behavior-agnostic interpretations. While valuable for general interpretability, they do not provide insights directly tied to specific model behaviors.

## 9 Discussion and conclusion

In this work, we introduced a framework for learning disentangled representations of the latent concepts that mediate a language model's behavior. We showed theoretically and empirically that when behavior changes are caused by sparse shifts in these mediating concepts, our proposed method accurately recovers features that align with the true underlying concepts. This sparsity assumption is motivated by identifiable representation learning approaches that leverage sparsity in the mappings from latent features to labels (Lachapelle et al., 2023) or transitions in temporal data (Lachapelle et al., 2022, 2024). While verifying whether such sparsity holds in a given dataset or model is challenging, the disentangled concepts that we found in the case study on refusal behavior of large language models suggest that the assumption is suited to naturally occurring data. We hope that our work can help bridge the gap between theoretical identifiability guarantees and practical interpretability in language models, demonstrating how assumptions like sparsity can yield meaningful and recoverable latent structure in real-world settings. Integrating different assumptions from the identifiability literature to expand the suite of weakly supervised LM interpretability tools and exploring the uses of our approach for steering are all avenues for future work.

## Acknowledgments

We would like to thank Kartik Ahuja and Sébastien Lachapelle for the very useful discussions and feedback on this work, especially the theoretical elements. This work was initiated at the Second Bellairs Workshop on Causality, held at the McGill University Bellairs Research Institute (January 6–13, 2023). This work was supported by the NSF Institute for Trustworthy AI in Law & Society (TRAILS; Award No. 2229885), NSERC Discovery Grant RGPIN-2023-04869, and a Canada CIFAR AI Chair. Any opinions, findings, or conclusions expressed in this material are those of the author(s) and do not necessarily reflect the views of the funding agencies.

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

# A Proofs

We repeat the Theorems 1 and 2 here for completeness.

**Theorem 1.** *If the Assumption 1 hold, then the function $\hat{\tau}$ that satisfies Equation 3 recovers the true mediating concepts $h(\mathbf{x})$ up to an invertible affine transformation, i.e., $\hat{f}(\mathbf{x}) = Ah(\mathbf{x}) + b$ for all $\mathbf{x}$.*

Proof. Let us write $\tau(g(\mathbf{x}))$ as $\hat{f}$, for simplicity. For a logistic linear predictor $\hat{h}$, the model family can be written as

$$p(c|\mathbf{x}) = \frac{\exp(\hat{f}(\mathbf{x})^T q(c))}{\sum_{j=1}^{m} \exp(\hat{f}(\mathbf{x})^T q(j))},$$

where $q(j) \in \mathbb{R}^d$ is the $j$-th column of the linear prediction matrix. Next, based on Assumption 1, the true posterior of the behavior group $c$ can be written as

$$p(c|\mathbf{z}) = \frac{\exp(\mathbf{w}_c^T \mathbf{z})}{\sum_{j=1}^{m} \exp(\mathbf{w}_j^T \mathbf{z})}.$$

The categorical cross entropy loss in objective Equation (3) is minimized when the optimal predictor $p(c|\hat{f}(\mathbf{x}))$ matches the posterior probability $p(c|\mathbf{z})$ for all the labels $c$ and so we can write

$$\frac{\exp(\hat{f}(\mathbf{x})^T q(c))}{\sum_{j=1}^{m} \exp(\hat{f}(\mathbf{x})^T q(j))} = \frac{\exp(\mathbf{w}_c^T \mathbf{z})}{\sum_{j=1}^{m} \exp(\mathbf{w}_j^T \mathbf{z})}.$$

Taking $\log$ on both sides, we get

$$\hat{f}(\mathbf{x})^T q(c) = \mathbf{w}_c^T \mathbf{z} + \mathbf{b}, \quad \forall c$$

where $\mathbf{b}$ is the difference of the normalization terms, and is independent of $c$. Writing

$$\hat{f}(\mathbf{x})^T q(0) = \mathbf{w}_0^T \mathbf{z} + \mathbf{b}$$
$$\hat{f}(\mathbf{x})^T q(1) = \mathbf{w}_1^T \mathbf{z} + \mathbf{b}$$
$$\vdots \qquad\qquad \vdots$$
$$\hat{f}(\mathbf{x})^T q(m) = \mathbf{w}_m^T \mathbf{z} + \mathbf{b}$$

we see that if $m > d$, we can write this as

$$\hat{f}(\mathbf{x})L = W\mathbf{z} + \mathbf{b}$$
$$\implies \hat{\mathbf{z}} = A\mathbf{z} + \mathbf{b}',$$

where the matrix $L$ is full-rank and invertible, and the last line follows from applying $L^{-1}$ on both sides. Since $W$ is also full-rank and invertible when $m > d$, the linear transformation $A$ is also invertible.

**Theorem 2.** *If Assumptions 1 and 2 hold, then the function $\hat{\tau}$ that satisfies Equations (3) and (4) identifies true latent up to permutation and scaling, i.e., $A = DP$ where $D$ is a diagonal scaling matrix and $P$ permutation matrix.*

Proof. To show this, we follow a similar proof as Ahuja et al. (2022a). Based on Assumption 2, we have for an $r \in \{1, 2, \ldots, d\}$, there exists some $C_k$ and $C_l$, such that $r$ is the causal differentiating factors between $C_k$ and $C_l$. From Corollary 1, we have $\mathbf{z}_{j*} = \mathbf{z}_{i*} + \boldsymbol{\delta}_{kl}$ for some $\mathbf{x}_{i*} \in C_k$ and $\mathbf{x}_{j*} \in C_l$, where $\boldsymbol{\delta}_{kl}$ is a $d$-dimensional vector which takes a nonzero value at position $r$ and $0$ elsewhere. Or simply $\mathbf{z}_{j*} = \mathbf{z}_{i*} + b_r\mathbf{e}_r$, where $\mathbf{e}_i = [0, \ldots, 1_i, \ldots, 0]$ is a $d$-dimensional identity vector with 1 at the $i^{th}$ position and 0 elsewhere and $b_r$ is a nonzero scalar.

For some pair of groups $C_k$ and $C_l$ that are related by a 1-sparse causal differentiating factor, the contrastive objective in Equation (4) yields a zero loss term if the learner correctly finds any of the inputs in the set corresponding to $i^*$ and $j^*$.

Suppose the learner guesses $\hat{\mathbf{z}}_i \in C_k$ and $\hat{\mathbf{z}}_{j^*} \in C_l$, such that $\hat{\mathbf{z}}_{j^*} = \hat{\mathbf{z}}_i + \hat{\boldsymbol{\delta}}_{kl}$, where $\hat{\boldsymbol{\delta}}_{kl}$ is the guessed perturbation with $\hat{\boldsymbol{\delta}}_{kl} = c_s \mathbf{e}_s$, such that $s \in \{1, \ldots, d\}$ and $c_s \neq 0$. That is, the learner guesses that the pair of points $\mathbf{x}_i$ and $\mathbf{x}_{j^*}$ satisfy the corollary. Our proof follows in two steps: first, considering the case when $\mathbf{x}_i$ is equivalent to $\mathbf{x}_{i^*}$, that is $\tau(g(\mathbf{x}_i)) = \tau(g(\mathbf{x}_{i^*}))$, or simply put $\mathbf{z}_i = \mathbf{z}_{i^*}$, and second, considering the case when $\mathbf{x}_i$ is **not** equivalent to $\mathbf{x}_{i^*}$.

Using Theorem 1, we have

$$\hat{\mathbf{z}}_{j^*} = A\mathbf{z}_{j^*} + \mathbf{b}$$
$$\hat{\mathbf{z}}_i + \hat{\boldsymbol{\delta}}_{kl} = A(\mathbf{z}_{i^*} + \boldsymbol{\delta}_{kl}) + \mathbf{b}$$

If $i \equiv i^*$, we can write

$$\hat{\mathbf{z}}_{i^*} + \hat{\boldsymbol{\delta}}_{kl} = A(\mathbf{z}_{i^*} + \boldsymbol{\delta}_{kl}) + \mathbf{b}$$
$$\hat{\mathbf{z}}_{i^*} + c_s \mathbf{e}_s = A(\mathbf{z}_{i^*} + b_r \mathbf{e}_r) + \mathbf{b}$$
$$\hat{\mathbf{z}}_{i^*} + c_s \mathbf{e}_s = A\mathbf{z}_{i^*} + b_r A\mathbf{e}_r + \mathbf{b}$$
$$c_s \mathbf{e}_s = b_r A\mathbf{e}_r$$
$$\frac{c_s}{b_r} \mathbf{e}_s = A\mathbf{e}_r \qquad (5)$$

This implies that the $r^{th}$ column of $A$ is $\frac{c_s}{b_r} \mathbf{e}_s$. This is because the $i^{th}$ entry on the right side would be the $i^{th}$ row of $A$ multiplied by $\mathbf{e}_r$. Since all values of $\mathbf{e}_r$, except the $r^{th}$ one, are zero, this multiplication would yield the $i^{th}$ entry on right side as $A_{ir}$. Therefore, $A_{ir}$ is zero for $i = 1, \ldots, d$, except $s$.

However, the learner can attempt to enforce the sufficient condition $\hat{\mathbf{z}}_{j^*} = \hat{\mathbf{z}}_i + \hat{\boldsymbol{\delta}}_{kl}$ for a different pair of points where $i \not\equiv i^*$ by learning a *axis-misaligned (i.e., linearly entangled)* solution for $\hat{\mathbf{z}} = \hat{\tau}(g(\mathbf{x}))$. In this case, when $i \not\equiv i^*$, the necessary condition in Assumption 2 will be violated. Essentially, if $\mathbf{x}_i \in C_k$, such that $\mathbf{z}_{j^*}$ and $\mathbf{z}_i$ differ along $\hat{\boldsymbol{\delta}}_{kl}$, which is not axis-aligned, we can write $\mathbf{z}_{j^*} = \mathbf{z}_j + c_u \mathbf{e}_u$, where $u \neq r$ (depicted in Figure 4 (left)). If this is the case, we get

$$\mathbf{z}_{i^*} = \mathbf{z}_{j^*} + b_r \mathbf{e}_r$$
$$\mathbf{z}_{i^*} = \mathbf{z}_j + b_r \mathbf{e}_r + c_u \mathbf{e}_u$$
$$A(\mathbf{z}_{i^*}) + \mathbf{b} = A(\mathbf{z}_j + b_r \mathbf{e}_r + c_u A\mathbf{e}_u) + \mathbf{b}$$
$$\hat{\mathbf{z}}_{i^*} = \hat{\mathbf{z}}_j + b_r A\mathbf{e}_r + c_u A\mathbf{e}_u$$

By the necessary condition, $\hat{\mathbf{z}}_{i^*}$ and $\hat{\mathbf{z}}_j$ should differ along at least $s$ (that is, the learned causal differentiating concept). However, if we choose $\mathbf{z}_j$, such that, $c_u = -b_r \frac{A_{sr}}{A_{su}}$, we get $\langle b_r A\mathbf{e}_r + c_u A\mathbf{e}_u, \mathbf{e}_s \rangle = 0$. That is, $\hat{\mathbf{z}_{i^*}}$ and $\hat{\mathbf{z}_j}$, which lead to different behaviors, differ by a factor that is *not* the *guessed* causal differentiating factor. This is a contradiction. Note that this contradiction does not happen if there is no such $\mathbf{z}_j \in C_l$. This case is shown in in Figure 4 (right), where the group $C_1$ does not contain the $\mathbf{z}_j$ point that could induce the contradiction. In this case, however, the model could use clusters $C_1$ and $C_3$ to learn the alignment for latent $z^2$ instead.

Now, since the span of both true and guessed perturbations is $d$, we get $d$ equations of the form Equation 5, such that for every $r$, there is a unique $j$. Note that the condition does not need to be met for all pairs of groups, but at least some pair of groups along a dimension $r \in \{1, \ldots, d\}$. Subsequently, applying the above argument to all column of $A$ yields $A$ as a permutation of an identity matrix. Note that, even though this condition is derived for pairs of groups that have 1-sparse causal differentiating concepts, since $\hat{\mathbf{z}} = A\mathbf{z} + \mathbf{b}, \forall \mathbf{z}$ (based on Theorem 1), if $A$ is permutation of identity matrix for some $\mathbf{z}$, it is true for all $\mathbf{z}$.

The argument in this section requires the learner to enforce sparse perturbations for pairs $C_k$ and $C_l$ that actually differ by a 1-sparse causal differentiating factor. However, the learner could cheat by attempting to enforce 1-sparse shifts between the wrong pair of groups. In the next section, we show that by encouraging sparse shifts across all pairs of groups, the objective in Equation (4) finds disentangled solutions.

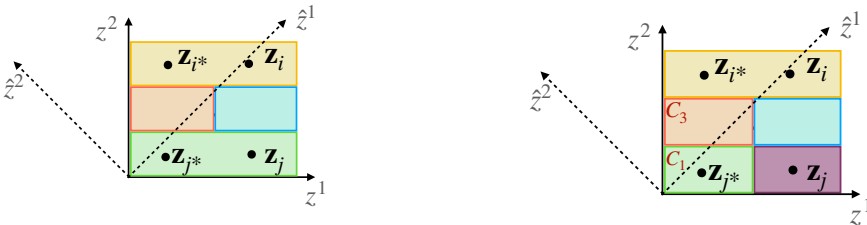

Figure 4: Visualization of the case when learner guesses causal differentiating concepts that is not axis-aligned.

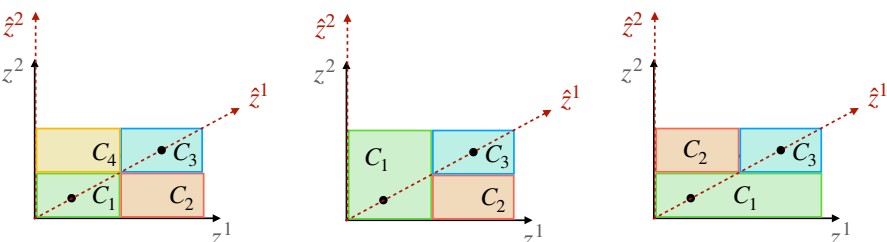

Figure 5: Base cases

**How does the constraint in Equation (4) enforce the correct alignment?** In the previous part, we show that if the learner enforces Equation (4) for some $C_k$ and $C_l$ such that $r$ is the 1-sparse causal differentiating concept between $C_k$ and $C_l$, we will have identifiability up to permutation and scaling. What remains is to argue that the optimization in (4) indeed drives the model to choose these correct cluster pairs. This is based on the argument that correctly aligned axes identify strictly more pairs of groups that satisfy the 1-sparse causal differentiating concept condition, and therefore yield a smaller loss in Equation (4). We formalize this intuition by induction on the number of groups along a given latent dimension.

**Setup.** Let us consider a case where all but one axis is misaligned, say $\hat{z}^1$ instead of $z^1$. We will show that the number of cluster pairs that have 1-sparse causal differentiating concept as the true latent $z^1$ is strictly larger than along the rotated axis $\hat{z}^1$. This argument then extends identically to the remaining $d-1$ dimensions.

For simplicity, we consider a two-dimensional latent space with the true coordinates $(z^1, z^2)$ and the learned latent space $(\hat{z}^1, z^2)$. We call a pair $C_k$ and $C_l$ a match-pair if it has a 1-sparse causal differentiating concept along the $z^1$. We want to show that there are strictly more match pairs along $z^1$ than along $\hat{z}^1$.

**Base cases.** We begin with the simplest case where there are only two distinct values along each axis. There are three possible configurations of groups, as illustrated in Figure 5 In the first case, we have two match pairs along the true axis $z^1$: $\{(C_1, C_2), (C_3, C_4)\}$, but only one match-pair along the rotated axis $\hat{z}^1$: $\{(C_1, C_3)\}$. Similarly, in the second case, we have two match pairs along the true axis $z^1$: $\{(C_1, C_2), (C_1, C_3)\}$, but again only one match-pair along the rotated axis $\hat{z}^1$: $\{(C_1, C_3)\}$. In the last case, however, we have only 1 match-pair along both the true and the rotated axes: $\{(C_2, C_3)\}$ and $(C_1, C_3)\}$, respectively. Thus, the strict inequality in the number of match pairs holds when there are more than two match pairs along $z^1$—that is, when the variation along $z^1$ is sufficient to reveal the structure of the causal differentiating concepts.

**Inductive step.** Next, we show that adding additional variation along the $z^1$ axis preserves this strict inequality. Consider the scenario depicted in Figure 6, where the shaded (yellow) region represents groupings already included up to iteration $i$. Let $n$ denote the number of discrete values taken by $z^2$, i.e., the number of values along the $z^2$ axis. Let $m_i$ denote the number of match pairs identified along the true axis $z^1$, and $m'_i$ denote the number of match pairs identified along the misaligned axis $\hat{z}^1$ up to iteration $i$. By the inductive hypothesis, $m_i > m'_i$. We want to show that $m_{i+1} > m'_{i+1}$.

When an additional group of set of groups are introduced along $z^1$, three cases can occur:

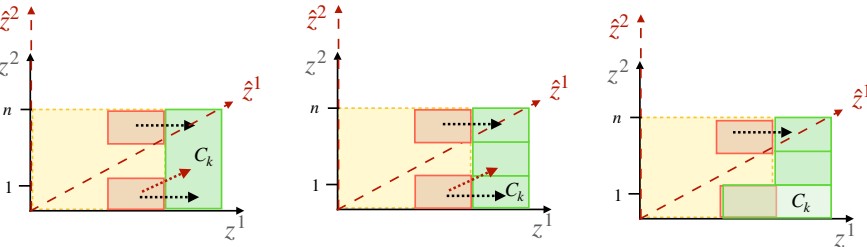

Figure 6: Induction cases

(1) *A new group $C_k$ is added.* This group forms a new match pair with at least one existing group for each of the $n$ possible values along $z^2$ (and possibly more, depending on the range along $z^1$). Hence, $m_{i+1} \geq m_i + n$. Along the misaligned axis $\hat{z}^1$, however, the same group can form at most $n - 1$ match pairs due to the rotation, so $m'_{i+1} \leq m'_i + n - 1$. Thus, $m_{i+1} - m'_{i+1} \geq (m_i - m'_i) + 1 \implies m_{i+1} \geq m'_{i+1}$.

(2) *Multiple new groups are added along $z^1$.* Each behaves analogously to case (1), and the inequality above holds again, so $m_{i+1} \geq m'_{i+1}$.

(3) *Merged groups.* One or more new observations may have similar behavior as the existing observations, leading to merged groups (rightmost case in Figure 6). In this case, no new match pairs are created or destroyed, so the inequality from the previous step is preserved.

Therefore, by induction, the number of match pairs along the true axis $z^1$ are strictly greater than along any misaligned axis $\hat{z}^1$. Since the objective in Equation (4) sums the contrastive constraint over all pairs of clusters, the alignment $\hat{z} = \hat{\tau}(g(\mathbf{x}))$ that yields more valid match pairs (i.e., the true disentangled alignment) achieves a strictly smaller objective value than any entangled alignment.

## B   Implementation details

**Data.**   For synthetic, we generate 20,000 samples. We consider the following mixing functions for linear, non-linear sparse, and non-linear non-sparse experiments:

$d = 2$

$$\mathbf{x} = [2z_0, 5z_1, z_0 + z_1, 2z_0 + z_1, -z_0 + 4z_1, 3z_0 - 2z_1] \qquad \text{(linear, non-sparse)}$$
$$\mathbf{x} = [z_0, 2z_0 + 3z_0^2, 4z_1, 2z_1^2 + z_1^3, 6\sin(z_0), -2\cos(z_1)] \qquad \text{(non-linear, sparse)}$$
$$\mathbf{x} = [z_0 z_1, z_0 + 3z_1, z_0^2 z_1, z_0 z_1^3, 2z_0 + z_1, 2z_0 z_1^2] \qquad \text{(non-linear, non-sparse)}$$

$d = 3$

$$\mathbf{x} = 2z_0, 5z_1, 3z_2, z_0 + z_1 + z_2, 2z_0 + z_1 + z_2, -z_0 + 4z_1 - 3z_2, 3z_0 - 2z_1 + 5z_2]$$
$$\mathbf{x} = [z_0, 2z_1, 6z_2, z_0 + 3z_0^2, z_1^2 + 4z_1^3, z_2 + 5z_2^2, z_0\cos(z_0), 6\sin(z_1), z_2\sin(z_2)]$$
$$\mathbf{x} = [z_0 z_1 z_2, z_0 + 3z_1 + 5z_2, z_0^2 z_1 z_2, z_0 z_1 z_2 \sin(z_2), 2z_0 + z_1 z_2, 2z_0 z_1^2 z_2^2]$$

For semi-synthetic data, we sample 20,000 bios from BiasBios dataset (De-Arteaga et al., 2019), which is available under Apache-2.0 license. We assign the high, medium, low occupation class based on the 1/3 and 2/3 quantile of median male salary for the respective occupation based on the U.S. Department of Labor's Employment and Earnings by Occupation statistics (https://www.dol.gov/agencies/wb/data/occupations).

For non-synthetic data, we use 500 examples each from harmful, harmless, and pseudo-harmful categories. Harmful examples are sampled uniformly from the MALICIOUSINSTRUCT (Huang et al. (2024), CC BY-SA-4.0 License), HARMBENCH (Mazeika et al. (2024), MIT License), ADVBENCH (Zou et al. (2023), MIT License), and TDC2023 (Mazeika et al. (2022), MIT License) datasets. Pseudo-harmful examples are sampled from OR-BENCH-80K (Cui et al. (2025), CC BY-4.0 License). All datasets follow a 70:15:15 train-validation-test split.

Table 3: Compute resources for different experiments. All runtimes were consistent across our method and the two baseline approaches.

| Data setting | Model | Compute resources | Approx. time |
|---|---|---|---|
| Synthetic | Feedforward NN | 4 CPU, 32 GB | 20 minutes |
| Semi-synthetic | DistilBert-base | 1 GPU, 4 CPU, 32 GB | 45 minutes |
| Semi-synthetic | Qwen2-7B | 4 GPU, 16 CPU, 256 GB | 17 hours |
| Semi-synthetic | Llama-3.1-8B | 4 GPU, 16 CPU, 256 GB | 17 hours |
| Non-synthetic | Qwen2-7B | 4 GPU, 16 CPU, 256 GB | 2.25 hours |
| Non-synthetic | Llama-3.1-8B | 4 GPU, 16 CPU, 256 GB | 2.25 hours |

**Models.**    In our experiments, the abstraction model $\mathcal{M}$ is implemented as a feed-forward network with ReLU activations. For synthetic data, the bottleneck consists of two linear layers, with a fixed hidden dimension of $4$, except for the final layer in the bottleneck, which has the output dimensionality of $d \in [2, 3]$. For semi-synthetic and non-synthetic data, the number of layers and dimensionality of the bottleneck is a hyperparameter (with $n_{\text{layers}} \in [2, 4, 8, 16]$ and $h_d \in [64, 128, 256, 512]$), with the hidden dimension of the final layers fixed to $d$. The predictor always consists of a single linear layer to match the loglinear assumption.

For the experiments with synthetic data, we estimate $g(\mathbf{x})$ with a feedforward neural network. Essentially, we construct a counterpart of the language model in Figure 1 by training an encoder-decoder model using a variational autoencoding objective. We then explain the causal factors in $g(\mathbf{x})$ by learning the alignment $\hat{\mathbf{z}} = \tau(g(\mathbf{x}))$ using the proposed method detailed in Section 3. The encoder-decoder model is implemented as a stack of 4 linear layers with ReLU activations, each with a hidden size of 16.

For the contrastive constraint, we estimate the causal differentiating factor $\boldsymbol{\delta}_{kl}$ by searching over the latent dimension $d$. Specifically, for a $\mathbf{x}_k \in C_k$, we sample $n = 5$ examples from another group $C_l$ and define the contrastive loss as $\min_j \mathbb{E}[||\hat{f}(\mathbf{x}_{lj}) - \hat{f}(\mathbf{x}_k) - \hat{\boldsymbol{\delta}}_{kl}||^2]$.

**Training.**    We use PyTorch[5] and HuggingFace Transformers[6] libraries for our experiments. For experiments with synthetic data, we train our models using the Adam optimizer and a learning rate scheduler that reduces the learning rate when the validation loss plateaus. The model is trained for 50 epochs and the best checkpoint is selected based on the validation loss.

For semi-synthetic and non-synthetic experiments, we use the default optimizer and scheduler provided in the Transformer training utils (AdamW and a linear learning rate scheduler). The model is trained for 3 epochs. In the semi-synthetic setting, the number and size of layers in the bottleneck modules are treated as hyperparameters, as detailed earlier. Hyperparameter selection is performed with grid search using the Ray Tune library[7] optimizing for disentanglement score on the validation dataset. For the non-synthetic dataset, where the ground-truth causal factors are not known, we reuse the best-performing hyperparameters identified in the semi-synthetic setting.

**Statistical analysis.**    For significance testing on experiments with synthetic and semi-synthetic data, we divide the test datasets into 5 splits. For each split, we compute the disentanglement, completeness, informativeness, and MCC scores. Under the assumption that the average scores are approximately normally distributed, we perform independent t-tests to assess statistical significance, with a threshold of **0.05**.

**Compute resources.**    The experiments in this paper were conducted on machines equipped with Tesla P100-PCIE-12GB GPUs. The resource usage along with compute time are shown in Table 3.

---

[5]https://pytorch.org/

[6]https://huggingface.co/docs/transformers/

[7]https://docs.ray.io/en/latest/index.html

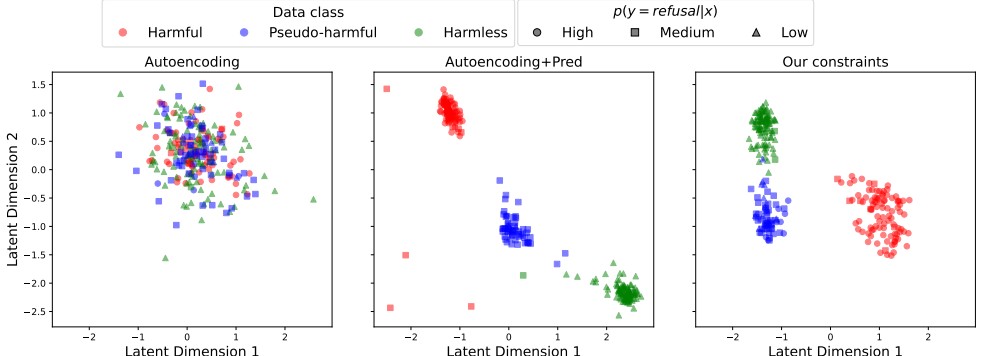

Figure 7: Latent space for refusal behavior in the Qwen2-7B model with autoencoding (*left*), autoencoding + prediction (*center*), and our constraints (*right*).

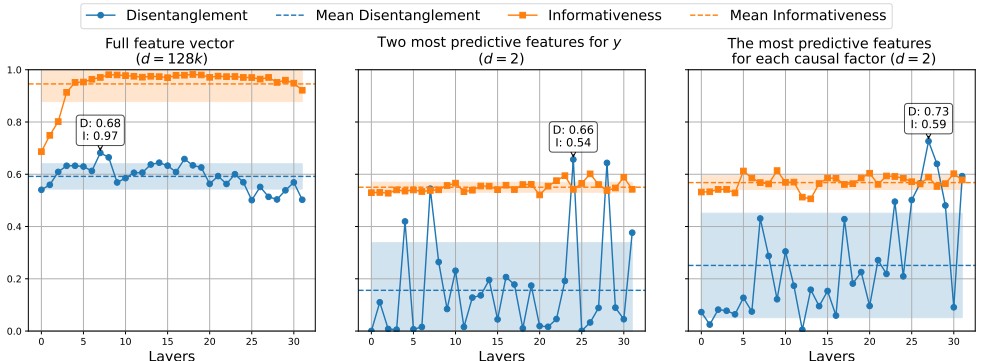

Figure 8: Disentanglement and informativeness scores for 128K-dimensional sparse autoencoders across different model layers. Solid lines represent the scores for each layer, with dashed lines indicating the mean, and shaded regions representing ±1 standard deviation. The maximum disentanglement score in each subplot is annotated along with its corresponding informativeness score.

## C   Additional results

**Case-study experiments on the Qwen2-7B model**   Figure 7 shows results on refusal behavior in language model on the Qwen2-7B model. Similar to the findings on the Llama-3.1-8B model (Figure 2), we see that the baseline method entangles the two latent dimensions, but the contrastive constraint leads to an aligned latent space with latent dimension 1 representing the harmfulness factor and the latent dimension 2 representing the topic factor. Notably, even though the model behavior is defined based on $p(\mathbf{y} = \text{refusal}|\mathbf{x})$, the learned latent space neatly separates the harmful, pseudo-harmful, and harmless prompts.

**Comparison to sparse autoencoder for 128k-dimensional SAE**   Figure 8 presents results on semi-synthetic dataset for the 128k-dimensional sparse autoencoder. The findings are consistent with those observed for the 32k model, as discussed in Section 6.

**Varying latent dimensions**   As discussed in Section 7, we perform experiments where we vary the dimensionality of the learned latents to assess how over-specification or under-specification affect disentanglement metrics. We consider two cases: (1) when the dimensionality of learned latents is **higher** than that of true mediating concepts, and (2) when the dimensionality of learned latents is **lower** than that of true mediating concepts.

In experiment 1, we construct synthetic data with the number of true mediating concepts set to 2 and the number of learned latent concepts set to 3 and 4. We find that if we relax the span constraints in Section 3, that is, the learned perturbations do not need to span all latent concepts, the learned

perturbations correspond to the true causal differentiating concepts. That is, the model disregards any extra latent(s) in learning the causal differentiating concepts, and the extra latents are solely some function of one of the other latent variables. Thus, the model achieves a high disentanglement score, but a lower completeness score (since more than one latent captures a true causal factor): DCI-D $0.95 \pm 0.03$ and DCI-C $0.81 \pm 0.14$ for $n = 3$, and DCI-D $0.94 \pm 0.04$ and DCI-C $0.76 \pm 0.19$ for $n = 4$. In sum, when the model's latent dimension is mis-specified to be larger than the true latent dimension, we observe a degree of robustness in the performance.

With the span constraint, the learned perturbations no longer correspond to the true causal differentiating concepts. This leads to degradation in disentanglement scores, alongside lower completeness scores: DCI-D $0.64 \pm 0.11$ and DCI-C $0.81 \pm 0.14$ for $n = 3$ and DCI-D $0.89 \pm 0.08$ and DCI-C $0.57 \pm 0.13$ for $n = 4$. Interestingly, the disentanglement scores are still higher than the autoencoding (DCI-D $0.19 \pm 0.12$, DCI-C $0.24 \pm 0.15$) and autoencoding + prediction (DCI-D $0.08 \pm 0.07$, DCI-C $0.09 \pm 0.08$) baselines. But more importantly, **overspecifying latent dimensionality consistently leads to a drop in completeness scores, which can be used as a signal for reducing the latent dimensionality as a hyperparameter.**

In experiment 2, we construct synthetic data with the number of true mediating concepts set to 3 and the number of learned latent concepts set to 2. We find that under-specifying the latent dimensionality results in a significant drop in the disentanglement scores ($0.51 \pm 0.20$). This is to be expected since the learned latents cannot separate out the mediating concepts since there are not enough latents. Thus, we conclude that **under-specifying latent dimensionality consistently leads to a lower disentanglement score, which can be used as a signal for increasing the latent dimensionality as a hyperparameter.**

**Choice of intervention layers**  To empirically test the impact of applying our approach to intermediate layers of LLMs instead of the last layer, we extend our refusal case study to intermediate representations and assess whether our method can extract relevant insights from intermediate LM layers as well. For simplicity, we consider the representation at the last token position in each layer of the Llama-3.1-8B model. We find that in the earlier layers, our approach does not yield disentangled representations. This is to be expected, as it has been noted before that initial LLM layers capture surface-level encodings of the inputs. However, past layer 9, the structure presented in Figure 2 starts to emerge. That is, we observe that the harmfulness and topicality dimensions identified in our experiments can also be disentangled in intermediate model representations. For a quantitative assessment, we calculate the average unsupervised disentanglement score between the model's internal representations and the last layer. We observe an unsupervised disentanglement score of $0.78 \pm 0.02$ for layers 10–30 compared to $0.39 \pm 0.20$ for the first 9 layers. Interestingly, these findings align with a recent paper (Zhao et al., 2025) that finds that harmfulness and refusal directions can be extracted from LLM representations for Llama2-Chat-7B around layer 10. It would be interesting for future exploration to study whether there are interesting concepts that can be extracted from intermediate layers that are abstracted away or lost in later layers. Our method, hopefully, provides a useful tool for such discoveries and insights.

**Violation of 1-sparse assumption**  Even though we restrict our discussion to 1-sparsity, we expect that our results can be extended to 2-sparse condition as well (or generally speaking, a p-sparse condition) following the proof strategy as in Ahuja et al. (2022a). That is, if the causal differentiating concepts are p-sparse and non-overlapping, we can expect identifiability up to permutations and block-diagonal transforms, rather than our stronger results on identifiability up to permutations and scaling. Further, if causal differentiating concepts are p-sparse and overlapping, we can expect that the latents at the intersection of these blocks will be identified up to permutations and scaling.

To validate this second case in our setting, we construct synthetic data with 2-sparse causal differentiating concepts for $d = 3$. Specifically, imagine three causal variables: occupation, gender, and experience that determine the income prediction behavior of the model, such that the $p(y|x)$ can be clustered into four classes: {(Doctors/Lawyers, Male, More experience), (Doctors/Lawyers, Female, Less experience); (Teachers/Nurses, Female, More experience), and (Teachers/Nurses, Male, Less experience)}. We see that in this data, groups $C_k$ and $C_l$ have have 2 causal differentiating concepts. For example, for behavior classes (Doctors/Lawyers, Male, More experience) and (Teachers/Nurses, Female, More experience), the causal differentiating concepts are occupation and gender.

We observe that our method can still disentangle the three causal factors. The average disentanglement measures (across 5 runs) are comparable to our 1-sparse setting—DCI-D ($0.95 \pm 0.02$), DCI-C ($0.98 \pm 0.01$), and DCI-I ($1.0 \pm 0.0$). Note that we get this disentanglement because the 2-sparse causal differentiating concepts are overlapping for different pairs of groups. This indicates that our method can be extended beyond the 1-sparse setting following the intuition from (Ahuja et al., 2022a). We leave more detailed explorations of these cases to future work.

