# OpenReview forum: "Causal Differentiating Concepts:  Interpreting LM Behavior via Causal Representation Learning"
_NeurIPS.cc/2025/Conference — NeurIPS 2025 spotlight_

### Official Review · Reviewer_mRsd · 2025-06-25

**Clarity:** 3
**Significance:** 3
**Originality:** 4
**Rating:** 4
**Confidence:** 4

**Summary:**

The paper introduces a method for discovering interpretable, behaviorally-relevant concepts from language model (LM) activations, called “causal differentiating concepts”. Causal differentiating concepts are defined as latent directions whose change are necessary and sufficient to alter the model’s output behavior. The method operates in an unsupervised manner, relying on a constrained contrastive learning objective and a key sparsity assumption: that behavioral differences between examples can be attributed to sparse shifts in latent concept space. The authors provide theoretical identifiability guarantees under these assumptions, and validate the method on synthetic, semi-synthetic, and real LLM-generated behaviors (e.g., refusal behavior in LLaMA-3.1–8B), showing superior disentanglement performance compared to sparse autoencoders and other baselines.

**Questions:**

a) Could the authors comment on whether applying the method to intermediate layers (rather than the final LLM layer) could yield more semantically meaningful concepts, even if less directly behavior-predictive?

b) How sensitive is the method to violations of the 1-sparse assumption? Have the authors tested scenarios with 2-sparse or distributed causal shifts?

c) In practice, how does one choose the number of latent dimensions? Is performance sensitive to this choice?

**Ethical Concerns:**

["NO or VERY MINOR ethics concerns only"]

**Final Justification:**

The authors addressed my questions. I am in favour of acceptence, however I am not revising my rating upwards, because I do not know how they intend to integrate their answers into the manuscript.

**Limitations:**

Yes

**Quality:**

3

**Strengths And Weaknesses:**

The paper is well-motivated and addresses an important challenge in interpretability: extracting concise, causal explanations of LLM behavior without relying on predefined labels or concepts. The formalization of "causal differentiating concepts" is elegant, and the method is both novel and theoretically grounded. The empirical evaluations are good.

However, there are two main concerns. First, the method extracts latent directions from the final layer of the LM, where representations may have already collapsed into task-specific, highly abstract forms. This design choice potentially limits the interpretability of the recovered concepts, since semantic richness is often better preserved in earlier layers. In practice, it is very likely that at least some of the differentiating concepts would collapse at the final layer, especially in more elaborate cases, where such interpretability methods would be most required.

---

> ### Author Rebuttal · Authors · 2025-07-31
>
> Thank you for your positive feedback on our work. We really appreciate your questions and comments on this work. To address your questions about the sensitivity of our method to specific modeling choices, we ran some additional experiments that evaluate misspecification of the latent dimension, violation of the 1-sparse assumption, and assess our ability to extract insights from additional layers of LMs.
>
> ---
>
> #### Applying the method to intermediate layers (rather than the final LLM layer)
> This is a great point. Indeed, intermediate layers of LLMs can often capture rich semantic information (possibly even richer than the last layer). Even though our experiments assess the last layer of LLMs, implementationally, our method can be directly applied to intermediate LLM layers too. Although our theoretical results do not directly translate to intermediate layers, a host of recent literature finds that latent concepts are linearly encoded in LM representations (linear representation hypothesis; Roeder et al. 2021; Jiang et al., 2024; Park et al., 2024). Since the contrastive learning constraints we develop in Eqn 4 are meant to disentangle linearly mixed concepts, we hypothesize that we can still use the method proposed in this paper for unsupervised discovery of unknown latent concepts present in the model’s internal representations.
>
> To empirically test this, we extend our refusal case study to intermediate representations to test whether our method can extract relevant insights from intermediate LM layers as well. For simplicity, we consider the representation at the last token position in each layer of the Llama-3.1-8B model. We find that in the earlier layers, our approach does not yield disentangled representations. This is to be expected, as it has been noted before that initial LLM layers capture surface-level encodings of the inputs. **However, past layer 9, the structure presented in Figure 2 starts to emerge. That is, we observe that the harmfulness and topicality dimensions identified in our experiments can also be disentangled in intermediate model representations.** For a quantitative assessment, we calculate the average unsupervised disentanglement score between the model's internal representations and the last layer. We observe an unsupervised disentanglement score of $0.78 \pm 0.02$ for layers 10-30 compared to $0.39 \pm 0.20$ for the first 9 layers. Interestingly, these findings align with a recent paper (Zhao et al., 2025), released earlier this month, that finds that harmfulness and refusal directions can be extracted from LLM representations for Llama2-Chat-7B around layer 10.
>
> It would be interesting for future exploration to study whether there are indeed interesting concepts that can be extracted from intermediate layers that are abstracted away or lost in later layers. Our method, hopefully, provides a useful tool for such discoveries and insights.
>
> ---
>
> #### Violations of the 1-sparse assumption
> This is a great question. Even though we restrict our discussion to 1-sparsity, we expect that our results can be extended to 2-sparse condition as well (or generally speaking, a p-sparse condition) following the proof strategy as Ahuja et al. (2022a). That is, if the causal differentiating concepts are p-sparse and non-overlapping, we can expect identifiability up to permutations and block-diagonal transforms, rather than our stronger results on identifiability up to permutations and scaling. Further, if causal differentiating concepts are p-sparse and overlapping, we expect that the latents at the intersection of these blocks to be identified up to permutations and scaling.
>
> To validate this second setting in our case, we construct synthetic data with 2-sparse causal differentiating concepts for latent dimension d=3. Specifically, imagine three causal variables: occupation, gender, and experience that determine the income prediction behavior of the model, such that the $p(y|x)$ can be clustered into four classes: {(Doctors/Lawyers, Male, More experience), (Doctors/Lawyers, Female, Less experience); (Teachers/Nurses, Female, More experience), and (Teachers/Nurses, Male, Less experience). We see that in this data, groups $C_k$ and $C_l$ have 2 causal differentiating concepts. For example, for behavior classes (Doctors/Lawyers, Male, More experience) and (Teachers/Nurses, Female, More experience), the causal differentiating concepts are occupation and gender.
>
> We observe that our method can still disentangle the three causal factors. The average disentanglement measures are comparable to our 1-sparse setting—DCI-D ($0.95 \pm 0.02$), DCI-C ($0.98 \pm 0.01$), and DCI-I ($1.0 \pm 0.0$). The reported numbers are across 5 runs with different initializations. Note that we get this disentanglement because the 2-sparse causal differentiating concepts are overlapping for different pairs of groups. **This indicates that our method can be extended beyond the 1-sparse setting following the intuition from Ahuja et al. (2022a).** However, we leave more detailed explorations of these cases to future work.
>
> Can you please clarify what you mean by “distributed causal shifts”? Does this mean that for any two behavioral classes, all (or multiple) latent concepts are causally differentiating? In this case, we would expect that our method would not yield disentangled representations since some form of inductive biases might be necessary for disentanglement. Please let us know if this does not address your question.
>
> ---
>
> #### Choice of latent dimensionality
> This is an interesting question. We conducted experiments to test variation in the learned number of latents in synthetic experiment settings and to assess factors that can help in choosing the latent dimension in practice.
>
> To this end, we consider two cases: (1) when the dimensionality of learned latents is **higher** than that of true mediating concepts, and (2) when the dimensionality of learned latents is **lower** than that of true mediating concepts.
>
> In experiment 1, we consider synthetic experiments with the number of true mediating concepts set to 2 and the number of learned latent concepts set to 3 and 4. We find that if we relax the span constraints (line 147), that is, the learned perturbations do not need to span all latent concepts, the learned perturbations correspond to the true causal differentiating concepts. That is, the model disregards any extra latent(s) in learning the causal differentiating concepts, and the extra latents are solely some function of one of the other latent variables. Thus, the model achieves a high disentanglement score, but a lower completeness score (since more than one latent captures a true causal factor): DCI-D $0.95 \pm 0.03$ and DCI-C $0.81 \pm 0.14$ for $n=3$, and DCI-D $0.94 \pm 0.04$ and DCI-C $0.76 \pm 0.19$ for $n=4$.  In sum, when the model's latent dimension is misspecified to be larger than the true latent dimension, we observe a degree of robustness in the performance.
>
> With the span constraint, the learned perturbations no longer correspond to the true causal differentiating concepts. This leads to degradation in disentanglement scores, alongside lower completeness scores: DCI-D $0.64 \pm 0.11$ and DCI-C $0.81 \pm 0.14$ for $n=3$ and DCI-D $0.89 \pm 0.08$ and DCI-C $0.57 \pm 0.13$ for $n=4$. Interestingly, the disentanglement scores are still higher than the autoencoding (DCI-D $0.19 \pm 0.12$, DCI-C $0.24 \pm 0.15$) and autoencoding + prediction (DCI-D $0.08 \pm 0.07$, DCI-C $0.09 \pm 0.08$) baselines. **But more importantly, overspecifying latent dimensionality consistently leads to a drop in completeness scores, which can be used as a signal for reducing the latent dimensionality as a hyperparameter.**
>
> For experiment 2, we consider synthetic experiments with the number of true mediating concepts set to 3 and the number of learned latent concepts set to 2. We find that underspecifying the latent dimensionality results in a significant drop in the disentanglement scores ($0.51 \pm 0.20$). This is to be expected since the learned latents cannot separate out the mediating concepts since there are not enough latents. **In this case, we conclude that underspecifying latent dimensionality consistently leads to a lower disentanglement score, which can be used as a signal for increasing the latent dimensionality as a hyperparameter.**
>
> We restrict ourselves to synthetic experiments for sensitivity analysis because in semi-synthetic and non-synthetic experiments, a parallel comparison is not feasible since the dimensionality of true mediating concepts is integral to the dataset and is not dictated externally. We believe that findings from synthetic experiments would naturally extend to general cases. Even though non-synthetic settings lack ground-truth mediating concepts, not allowing direct evaluation of disentanglement and completeness scores, the disentanglement measures can be measured across multiple model runs as a signal for guiding the latent dimensionality, following Duan et al. (2021).
>
> ---
>
> #### References
> [1] Kartik Ahuja, Jason S Hartford, and Yoshua Bengio. Weakly supervised representation learning with sparse perturbations. NeurIPS 2022.
> [2] Sunny Duan, Loic Matthey, Andre Saraiva, Nick Watters, Chris Burgess, Alexander Lerchner, Irina Higgins. Unsupervised Model Selection for Variational Disentangled Representation Learning. ICLR 2020.
> [3] Yibo Jiang, Goutham Rajendran, Pradeep Ravikumar, Bryon Aragam, Victor Veitch. On the Origins of Linear Representations in Large Language Models. ICML 2024.
> [4] Kiho Park, Yo Joong Choe, Victor Veitch. The linear representation hypothesis and the geometry of large language models. ICML 2024.
> [5] Geoffrey Roeder, Luke Metz, Diederik P. Kingma. On Linear Identifiability of Learned Representations. ICML 2021.
> [6] Jiachen Zhao, Jing Huang, Zhengxuan Wu, David Bau, Weiyan Shi. LLMs Encode Harmfulness and Refusal Separately. 2025.

---

> > ### Author Response · Authors · 2025-08-06
> >
> > Thank you again for your feedback on our work. We hope that our rebuttal has helped alleviate your concern about the applicability of our method in internal representations of LLMs. We have also included additional discussion and experiments for violations of 1-sparse assumptions and choice of latent dimensionality. Please let us know if these experiments have appropriately addressed your questions and concerns. We hope that you will factor these in your final assessment. We are open to any further questions.

---

> > ### Comment · Reviewer_mRsd · 2025-08-06
> > **Thank you**
> >
> > Thank you for the additioanl answering my questions. I have no further follow up questions.

---

### Official Review · Reviewer_D9ui · 2025-07-03

**Clarity:** 3
**Significance:** 1
**Originality:** 1
**Rating:** 3
**Confidence:** 3

**Summary:**

This paper studies the problem of extracting interpretable variables from LLM activations, under various assumptions. Given a dataset of paired texts x and discrete labels c (coming from an LLM), it's an important problem to be able to extract out the concepts that are responsible for the labels (assuming such a structure exists). The work considers the setting that there exist a set of mediating concepts z that are a function tau(g(x)) of the LLM activations g(x). Furthermore, the work assumes
1. z is mediating, i.e. c is indepenent of x given z
2. c is related to z via a logit-linear function.
3. Every concept is a 1-sparse causal differentiating concept: There exist a pair of classes such that changing this concept is necessary and sufficient to change from one class to the other.

Under these assumptions, it's shown that z is identifiable up to linear transformations. Moreover, the authors propose a simple loss function to learn these latents, which is a variant of the contrastive loss. Experiments on synthetic data, BiasBios data and LLM text data show the effectiveness of their approach, compared to some prior works such as SAEs, when we look at disentanglement and MCC metrics. The target audience are people interested in interpretability of LLMs.

**Questions:**

Some questions were raised above.

- It seems like for the proof to go through, it maybe possible to replace 1-sparsity to some sort of "full rank" condition on concepts <-> label structure. Could the authors comment on this?

**Ethical Concerns:**

["NO or VERY MINOR ethics concerns only"]

**Final Justification:**

While the authors have attempted to clarify my questions, the issues I raised are not fully resolved. The authors agree with the weaknesses of the paper and try to justify them with various reasons (which are not too convincing and are rather circumstantial). Therefore, I decide to keep my rating, however I will not oppose of majority of the reviewers like the work and wish to accept the work.

**Limitations:**

Limitations have been discussed.

**Quality:**

2

**Strengths And Weaknesses:**

## Strengths:

- The paper approaches the interesting and relevant problem of extract interpretable variables from LLM representations. This is important for many reasons, such as to figure out inherent biases in LLM systems.

## Weaknesses:

- The assumptions seem a bit too strong. For instance, assuming that every relevant concept for a given set of labels is a 1-sparse causal differentiating concept is a very unique structure that is unlikely to hold in practice. It seems like the proof or technique may break if we have 2-sparsity (for an appropriate definition) instead of 1-sparsity. Could the authors comment on this?

- The assumption that the mediating concepts are linearly related to the output variables is also very strong, since it essentially abstracts away the nonlinearity and turns the problem into a standard linear regression setting.

- The theorems are fairly straightforward given the strong assumptions, e.g. theorem 1 follows by simple manipulations once linearity is assumed and theorem 2 essentially follows the proof ideas from Ahuja et al. 2022, so theoretical novelty is quite limited.

- The number labels studied in the experiments seem very small, compared to what the approach or method can handle. It would be nice to have experiments on slightly more complicated real-world datasets with multiple classes.

---

> ### Author Rebuttal · Authors · 2025-07-31
>
> Thank you for your feedback on our work. We share some clarifications on the validity of the assumptions and the applicability of our method, which will hopefully help address your concerns.
>
> ---
>
> #### Would the proof or technique break if we have 2-sparsity instead of 1-sparsity?
> Even though we restrict our discussion to 1-sparsity, we expect that our results can be extended to 2-sparse condition as well (or generally speaking, a p-sparse condition) following the proof strategy as Ahuja et al. (2022a). That is, if the causal differentiating concepts are p-sparse and non-overlapping, we can expect identifiability up to permutations and block-diagonal transforms, rather than our stronger results on identifiability up to permutations and scaling. Further, if causal differentiating concepts are p-sparse and overlapping, we expect that the latents at the intersection of these blocks to be identified up to permutations and scaling.
>
> To validate this second setting in our case, we construct synthetic data with 2-sparse causal differentiating concepts for d=3. Specifically, imagine three causal variables: occupation, gender, and experience that determine the income prediction behavior of the model, such that the $p(y|x)$ can be clustered into four classes: {(Doctors/Lawyers, Male, More experience), (Doctors/Lawyers, Female, Less experience); (Teachers/Nurses, Female, More experience), and (Teachers/Nurses, Male, Less experience). We see that in this data, groups $C_k$ and $C_l$ have 2 causal differentiating concepts. For example, for behavior classes (Doctors/Lawyers, Male, More experience) and (Teachers/Nurses, Female, More experience), the causal differentiating concepts are occupation and gender.
>
> We observe that our method can still disentangle the three causal factors. The average disentanglement measures (across 5 initializations) are comparable to our 1-sparse setting—DCI-D ($0.95 \pm 0.02$), DCI-C ($0.98 \pm 0.01$), and DCI-I ($1.0 \pm 0.0$). This indicates that our method can be extended beyond the 1-sparse setting as well.
>
> With respect to the feasibility of 1-sparse assumption, we want to note that our 1-sparse assumption is more relaxed than that of Ahuja et al. (2022a) since we do not require that every pair of clusters have only 1 causal differentiating concept or that the pair of clusters that have 1 causal differentiating concepts are known beforehand, rather that every $z^r$ is 1-sparse for some pair of clusters. This assumption is easier to meet in practice, as unless some points in input space are not observed altogether, 1-sparsity would hold naturally. For instance, in the above example with 2-sparsity, we do not observe data points that are {Doctor/Lawyer, Female, More experience} at all. If we did, we can have one of following cases: (a) this is a separate behavior cluster, in which case this cluster and (Doctors/Lawyers, Male, More experience) will have gender as the causal differentiating concept and this cluster and (Teachers/Nurses, Female, More experience) will have occupation as the causal differentiating concept; (b) we have {Doctor/Lawyer, Female} cluster (including both more and less experience data points), in which case this and (Teachers/Nurses, Female, More experience) will still have occupation as the only causal differentiating concept; and so on. We hope this example helps convey why 1-sparsity is not a particularly strong assumption in practice, as long as sufficient variability in the input space is observed.
>
> ---
>
> #### Linearity assumption
> Log-linearity prediction is ubiquitous not only in causal representation learning literature (e.g., Hyvärinen and Morioka, 2016; Ahuja et al., 2022b) but also in machine learning. The idea is that the non-linearity in data is pushed into the feature extraction, and the outcome (e.g., next-token, label, etc.) is predicted linearly from the extracted features. Thus, the expressivity of such a setup is not akin to linear regression.
>
>
> While log-linearity is certainly an assumption, previous work in CRL theoretically proves that unsupervised learning of disentangled representations is fundamentally impossible without inductive biases (Locatello et al., 2019). Thus, certain assumptions are inherent to the goal of disentanglement. In the context of LLMs, the log-linearity assumption may not be as limiting since a host of recent work has found that concepts are linearly encoded internally (linear representation hypothesis; Roeder et al. 2021; Jiang et al., 2024; Park et al., 2024), lending themselves to be discovered by our method without the need of supervision.
>
> ---
>
> #### Theoretical novelty
> The judgment of novelty is inherently subjective; however, we would like to add that our paper does not claim or promise novelty in proof technique (our proof sketch clearly lays out that the proof naturally follows using techniques from Ahuja et al. (2022b), Hyvärinen and Morioka (2016), and Ahuja et al. (2022a)). A lot of mathematical work, generally, and especially in CRL, focuses on using existing proof techniques to prove theorems in new settings for new insights. The goal of our paper is to develop novel assumptions that take LLM behavior into account, enabling us to use the CRL framework to interpret LLMs without the need for labeled examples, a setting which is majorly under-explored in causal representation learning, with the exception of few recent works (Rajendran et al., 2024; Ahuja et al. 2022b; Moran and Aragam, 2025).
>
> Another thing to note is that the assumptions that we develop are novel. Reiterating an earlier point, our 1-sparsity assumption is more relaxed than that of Ahuja et al. (2022a) since we do not require that every pair of clusters have only 1 causal differentiating concept or that the pair of clusters that have 1 causal differentiating concept are known beforehand. Thus, the assumptions and proofs from previous work do not out-of-the-box translate to our setting. Rather, we identify more relaxed assumptions that can still use existing proof techniques for identifiable discovery of latent concepts.
>
> ---
>
> #### Number of labels
> Can you clarify why you believe that increasing the number of classes (presumably for the outcome y) could change the conclusion of our experiments, where we already consider behaviors that are non-binary? As we note in the paper, behaviors can be finer-grained than the number of classes, e.g., P(refuse|prompt) can be very high, high, medium, etc. Indeed, in the real-world experiments, we consider behaviors that are continuous valued and bucketed into 3 behavioral classes.
>
> In terms of comparison with the scope of other interpretability literature, a host of interpretability work spanning SAEs, mechanistic interpretability, mediation analysis, and more study binarized setting, for example, refusal behavior in language models (whether the model refuses a query or not; e.g., Zhao et al., 2025, Yeo et al. 2025), truthfulness in LLMs (whether model output is truthful or not; e.g., Zou et al., 2025), subject-verb agreement (whether subject and verb agree or not; e.g., Marks et al. 2025) and so on. In any of these works and ours, it is not that we restrict the output space of LLMs—they can still produce free-form text—but we interpret a property of this free-form text (refusal, truthfulness, etc.). Thus, our method, similar to existing approaches, can analyze LLMs out-of-the-box, be it a pretrained model, an instruction-tuned model, or a finetuned model.
>
> ---
>
>
> #### Replacing 1-sparsity with some “full rank” condition on concepts <-> label structure
> Just to clarify, by “full rank condition on concepts <-> label structure”, did you mean that if the perturbation matrix $\Delta$, which is comprised of the latent concepts that differentiate some labels $c_k$ and $c_l$, is full rank? Based on that interpretation, we do not think full rank condition guarantees identifiability. Consider $\Delta$ as the set of true perturbations and $\Delta’$ as the set of guessed perturbations. Theorem 1 yields $\Delta’ = A \Delta$, where $A$ is an affine transformation matrix. Now, if we assume that $\Delta$ is full rank, we can conclude that $A$ would be invertible (since both $\Delta$ and $\Delta’$ have rank n, so would the matrix $A$). But, it does not say anything about $A$ being an identity matrix (or a permutation of an identity matrix). A similar argument appears in Ahuja et al. (2022a) to prove that full rankness of $\Delta$ implies that true latents can be identified up to affine transformation, but for identifiability up to permutation and scaling, they use additional sparsity conditions as well.
>
> ---
>
> #### References
> [1] Kartik Ahuja, Jason S Hartford, and Yoshua Bengio. Weakly supervised representation learning with sparse perturbations. NeurIPS 2022.
> [2] Kartik Ahuja, Divyat Mahajan, Vasilis Syrgkanis, Ioannis Mitliagkas. Towards efficient representation identification in supervised learning. CLeaR 2022.
> [3] Aapo Hyvärinen, Hiroshi Morioka. Unsupervised Feature Extraction by Time-Contrastive Learning and Nonlinear ICA. NeurIPS 2016.
> [4] Samuel Marks, Can Rager, Eric J. Michaud, Yonatan Belinkov, David Bau, Aaron Mueller. Sparse Feature Circuits: Discovering and Editing Interpretable Causal Graphs in Language Models. ICLR 2025.
> [5] Gemma E. Moran, Bryon Aragam. Towards Interpretable Deep Generative Models via Causal Representation Learning. 2025.
> [6] Goutham Rajendran, Simon Buchholz, Bryon Aragam, Bernhard Schölkopf, Pradeep Ravikumar. From Causal to Concept-Based Representation Learning. NeurIPS 2024.
> [7] Wei Jie Yeo, Nirmalendu Prakash, Clement Neo, Roy Ka-Wei Lee, Erik Cambria, Ranjan Satapathy. Understanding Refusal in Language Models with Sparse Autoencoders. 2025.
> [8] Jiachen Zhao, Jing Huang, Zhengxuan Wu, David Bau, Weiyan Shi. LLMs Encode Harmfulness and Refusal Separately. 2025.
> [9] Andy Zou et al. Representation Engineering: A Top-Down Approach to AI Transparency. 2025.

---

> > ### Comment · Reviewer_D9ui · 2025-08-05
> >
> > I thank the authors for their detailed response.
> >
> > > if the causal differentiating concepts are p-sparse and non-overlapping, we can expect identifiability
> >
> > While I agree with this claim at a high-level, the difficulty indeed lies in the details of how to establish this rigorously. I thank the authors for the additional experiments that suggests something along these lines should be true, however I still suspect that showing it formally will require additional work.
> >
> > > Log-linearity prediction is ubiquitous not only in causal representation learning literature but also in machine learning
> >
> > My issue is not in the form of log-linearity which indeed occurs in multiple places in ML but in the assumption that z and c are log-linearly related, especially since the definition of z is abstracted away with the presence of an encoder \tau.
> >
> > > In the context of LLMs, the log-linearity assumption may not be as limiting since a host of recent work has found that concepts are linearly encoded internally (linear representation hypothesis; Roeder et al. 2021; Jiang et al., 2024; Park et al., 2024)
> >
> > The linear representation hypothesis (as stated in various works) does not include the presence of the nonlinear function \tau that this work uses. Therefore, this may not be a valid justification of the linearity assumption.
> >
> > > Can you clarify why you believe that increasing the number of classes (presumably for the outcome y) could change the conclusion of our experiments, where we already consider behaviors that are non-binary?
> >
> > To clarify, I'm saying the authors could have additionally done experiments with a larger d, to the probe the validity of the algorithm. This is especially possible in the synthetic setting.
> >
> > > Just to clarify, by “full rank condition on concepts <-> label structure”, did you mean that if the perturbation matrix ... is full rank?
> >
> > Yes, thanks for the comment. I agree that it may not be that straightforward to extend this result as your counterexample shows.

---

> > ### Author Response · Authors · 2025-08-06
> >
> > Thank you for engaging with our rebuttal. We wanted to clarify a couple of things raised in your response.
> >
> > ---
> >
> > Log-linearity assumption and linear representation hypothesis
> >
> > We'd like to clarify why we brought up the linear representation hypothesis (LRH). To start, note that to prove disentanglement, we follow a large body of work in CRL in first showing that representations can be identified up to linear transformations (our Theorem 1) and then showing that with assumptions, this indeterminacy can be further resolved up to permutations and scaling (Theorem 2). For the linear identifiability step in Theorem 1, we follow multiple papers in CRL (e.g., Ahuja et al. 2022a, Roeder et al. 2020, Hyvarinen and Morioka, 2016) in leveraging an assumption that the true latent concepts $z$ of interest are log-linearly related to behaviors $c$. Our point in bringing up the LRH was to highlight an alternative route to showing the disentanglement result of Theorem 2: the starting point could instead be to assume the LRH, that concepts are already linearly encoded by LM representations after multiple layers of nonlinearity. We can then directly prove Theorem 2 by leveraging a linear unmixing function $\tau$.
> >
> > This can be seen from additional experiments that we conducted, where we applied our method to intermediate language model layers. Please refer to the rebuttal to reviewer txCi “Applying the method to intermediate language model layers”. We find that even intermediate representations (past layer 9 in Llma-3.1-8B model) that are in practice non-linearly related to the outcome can be used to extract the disentangled concepts in our refusal case study.
> >
> > ---
> >
> > Formal proofs for p-sparse causal differentiating concepts
> > We agree with the reviewer that both intuitively and experimentally, we can expect our disentanglement results to extend to a p-sparse setting. However, as the reviewer pointed out, formally establishing p-sparse results would require more discussion. We will add a discussion of these intuitions and formalizations in the final version of the paper.
> >
> > Nevertheless, we would like to highlight that even without these additional theoretical results, our paper makes important and interesting contributions by proposing a novel method for unsupervised discovery of latent concepts that drive language model behavior. This is a novel and underexplored setting for causal representation learning. We theoretically and empirically show the ability of our method to recover ground-truth causal factors and further demonstrate how this method can be applied in the wild for interpreting LLM behavior. Our paper proposes novel assumptions that can leverage existing proof techniques and be applied to new settings.
> >
> > ---
> >
> > To summarize, we hope that our discussion here has been sufficient to alleviate the reviewer’s concern regarding the limitations of the 1-sparse causal differentiating assumption or the log-linearity assumption. As we have showcased here,
> >
> > 1. Our 1-sparse assumption is not as strict as it may seem. As seen from our example with a 2-sparse structure, if there is sufficient variability in the observation space, 1-sparsity is likely to hold. It is, of course, impossible to ascertain that every latent concept of interest can be identified using our method across all possible data-generating processes, but given the unsupervised nature of our method, it could be a useful toolset to discover interesting concepts that affect LLM behavior.
> >
> > 2. Our log-linearity assumption is also not as limiting, since many concepts are linearly encoded in LLM representations and thus, can be identifiably recovered using Theorem 2 even without the log-linear assumption.
> >
> > Please let us know if you have any further questions. We hope that you will factor these discussions in your assessment of our work.

---

### Official Review · Reviewer_txCi · 2025-07-03

**Clarity:** 3
**Significance:** 3
**Originality:** 4
**Rating:** 6
**Confidence:** 4

**Summary:**

This paper introduces a novel causal representation learning object which aims to recover the latent factors underlying model behaviour using an labelled dataset. Unlike previous approaches (e.g. causal abstract), this method does not need the hypothesis about the latent factors up front, but rather it uses key properties of the data and assumptions about the latent variables to learn the factors by itself. The authors provide results on both synthetic and real data which validates their method against alternatives without their causal objective, and baselines like sparse autoencoders.

**Questions:**

- The bottleneck is currently applied to the last layer hidden representations as justified by assumption 1. Is there any setup under which this method could be applied to internal representations, or does assumption 1 render this unjustifiable? This would enable identifying latent factors in model-internal components, which is useful for interpretability. It would also enable an alternative to SAEs.
- If the dimensionality of the bottleneck doesn't correspond well to the true number of factors, what happens? E.g. if only 1 dimension or 5 dimensions are given to the harmfulness task, are the learned factors still cleanly basis-aligned? I'd be curious if it results in interesting decomposition of the topic factor (latent 2 in figure 2). Some experiments or at least theoretical analysis of this would be informative.

**Ethical Concerns:**

["NO or VERY MINOR ethics concerns only"]

**Final Justification:**

All questions satisfied with additional experiments, no further questions or objections remain. The method seems solid; future work can test it on more real-world datasets.

**Limitations:**

yes

**Quality:**

4

**Strengths And Weaknesses:**

Strengths
- The paper is very mathematically clean and offers a new approach to a real thorny issue in causal abstraction: generating hypotheses about underlying latent variables in the causal model is currently a fully manual process. If we could learn the latent variables that would make causal abstraction far more scalable, which this work seems to make progress towards doing.
- The experiments involve several levels of complexity (from clean synthetic data to a more complex collection of real-world data on a relevant topic: refusal behaviour) and compare against well-formed baselines. It is great to see the inclusion of currently practical and in-vogue methods like SAEs. Outperforming these baselines builds convincing evidence for the validity of the method.

Weaknesses
- The real-world dataset is quite cleanly organised. It would be nice if this method could be applied to far messier real-world data like the baseline SAEs that were compared against. But this is not a significant weakness since this may require a wholly new method; this approach seems suitable for the problem the authors put in the scope of the paper.

---

> ### Author Rebuttal · Authors · 2025-07-31
>
> Thank you for your positive feedback on the theory and experiments presented in our work. We appreciate your recognition of the value of our work in scaling causal abstractions through unsupervised discovery of high-level concepts that affect model behavior. We also appreciate your positive comments regarding our experiments and evaluation. To address the questions raised about the sensitivity to variation in some modeling choices, we ran 3 additional experiments that evaluate misspecification of the latent dimension and assess our ability to extract insights from intermediate layers of LMs.
>
> ---
>
> #### Applying the method to intermediate language model layers
> This is a great point. In our experiments, we restrict ourselves to interpreting the last layer of the language models; however, experimentally, our method can be directly applied to intermediate LM layers as well. Although our theoretical results do not directly translate to intermediate representations, a host of recent literature identifies that latent concepts are linearly encoded in LM representations (linear representation hypothesis; Roeder et al., 2021; Jiang et al., 2024; Park et al., 2024). Since the contrastive learning constraints we develop in Eqn 4 are meant to disentangle linearly mixed concepts, we hypothesize that we can still use the method proposed in this paper for unsupervised discovery of unknown latent concepts present in the model’s internal representations.
>
> To empirically test this, we extend our refusal case study to intermediate representations to test whether our method can extract relevant insights from intermediate LM layers as well. For simplicity, we consider the representation at the last token position in each layer of the Llama-3.1-8B model, which is a common practice in other work studying internal LLM representations (Zou et al., 2022; Arditi et al., 2024). We find that in the earlier layers, our approach does not yield disentangled representations. This is to be expected, as it has been noted before that initial LLM layers capture surface-level encodings of the inputs. **However, past layer 9, the structure presented in Figure 2 starts to emerge. That is, we observe that the harmfulness and topicality dimensions identified in our experiments can also be disentangled in intermediate model representations.** For a quantitative assessment, we calculate the average unsupervised disentanglement score between the model's internal representations and the last layer. We observe an unsupervised disentanglement score of $0.78 \pm 0.02$ for layers 10-30 compared to $0.39 \pm 0.20$ for the first 9 layers. Interestingly, these findings align with a recent paper (Zhao et al., 2025), released earlier this month, that finds that harmfulness and refusal directions can be extracted from LLM representations for Llama2-Chat-7B around layer 10.
>
> ---
>
> #### If the dimensionality of the bottleneck doesn't correspond well to the true number of factors, what happens?
> This is an interesting question. We conducted experiments to test variation in the learned number of latents in synthetic experiment settings. We consider two cases: (1) when the dimensionality of learned latents is **higher** than that of true mediating concepts, and (2) when the dimensionality of learned latents is **lower** than that of true mediating concepts.
>
> In experiment 1, we consider synthetic data with the number of true mediating concepts set to 2 and the number of learned latent concepts set to 3 and 4. We find that if we relax the span constraints (line 147), that is, the learned perturbations do not need to span all latent concepts, the learned perturbations correspond to the true causal differentiating concepts. That is, the model disregards any extra latent(s) in learning the causal differentiating concepts, and the extra latents are solely some function of one of the other latent variables. Thus, the model achieves a high disentanglement score, but a lower completeness score (since more than one latent captures a true causal factor): DCI-D $0.95 \pm 0.03$ and DCI-C $0.81 \pm 0.14$ for $n=3$, and DCI-D $0.94 \pm 0.04$ and DCI-C $0.76 \pm 0.19$ for $n=4$.  In sum, when the model's latent dimension is misspecified to be larger than the true latent dimension, we observe a degree of robustness in the performance.
>
> With the span constraint, the learned perturbations no longer correspond to the true causal differentiating concepts. This leads to degradation in disentanglement scores, alongside lower completeness scores: DCI-D $0.64 \pm 0.11$ and DCI-C $0.81 \pm 0.14$ for $n=3$ and DCI-D $0.89 \pm 0.08$ and DCI-C $0.57 \pm 0.13$ for $n=4$. Interestingly, the disentanglement scores are still higher than the autoencoding (DCI-D $0.19 \pm 0.12$, DCI-C $0.24 \pm 0.15$) and autoencoding + prediction (DCI-D $0.08 \pm 0.07$, DCI-C $0.09 \pm 0.08$) baselines. **But more importantly, overspecifying latent dimensionality consistently leads to a drop in completeness scores, which can be used as a signal for reducing the latent dimensionality as a hyperparameter.**
>
> For experiment 2, we consider synthetic data with the number of true mediating concepts set to 3 and the number of learned latent concepts set to 2. We find that underspecifying the latent dimensionality results in a significant drop in the disentanglement scores ($0.51 \pm 0.20$). This is to be expected since the learned latents cannot separate out the mediating concepts since there are not enough latents. **In this case, we conclude that underspecifying latent dimensionality consistently leads to a lower disentanglement score, which can be used as a signal for increasing the latent dimensionality as a hyperparameter.**
>
> We restrict ourselves to synthetic experiments for sensitivity analysis because in semi-synthetic and non-synthetic experiments, a parallel comparison is not feasible since the dimensionality of true mediating concepts is integral to the dataset and is not dictated externally. We believe that findings from synthetic experiments would naturally extend to general cases. Even though non-synthetic settings lack ground-truth mediating concepts, not allowing direct evaluation of disentanglement and completeness scores, the disentanglement measures can be measured across multiple model runs as a signal for guiding the latent dimensionality, following Duan et al. (2021).
>
> ---
>
> #### References
> [1] Andy Arditi, Oscar Obeso, Aaquib Syed, Daniel Paleka, Nina Panickssery, Wes Gurnee, Neel Nanda. Refusal in Language Models Is Mediated by a Single Direction. NeurIPS 2025.
> [2] Sunny Duan, Loic Matthey, Andre Saraiva, Nick Watters, Chris Burgess, Alexander Lerchner, Irina Higgins. Unsupervised Model Selection for Variational Disentangled Representation Learning. ICLR 2020.
> [3] Yibo Jiang, Goutham Rajendran, Pradeep Ravikumar, Bryon Aragam, Victor Veitch. On the Origins of Linear Representations in Large Language Models. ICML 2024.
> [4] Kiho Park, Yo Joong Choe, Victor Veitch. The linear representation hypothesis and the geometry of large language models. ICML 2024.
> [5] Geoffrey Roeder, Luke Metz, Diederik P. Kingma. On Linear Identifiability of Learned Representations. ICML 2021.
> [6] Jiachen Zhao, Jing Huang, Zhengxuan Wu, David Bau, Weiyan Shi. LLMs Encode Harmfulness and Refusal Separately. 2025.
> [7] Andy Zou et al. Representation Engineering: A Top-Down Approach to AI Transparency. 2025.

---

### Official Review · Reviewer_2QmS · 2025-07-03

**Clarity:** 3
**Significance:** 3
**Originality:** 3
**Rating:** 5
**Confidence:** 2

**Summary:**

The authors claim 3 contributions:
1. "Causal differentiating concepts" - factors that much change to eliciting different model behavior - and a sparsity assumption for their identification.
2. A contrastive learning algorithm to identify the above concepts which is proven recover disentangled features.
3. Synthetic experiments and a real-world case-study supporting the above results.

**Questions:**

1. Is there evidence to support the claim that the true conditional probability distribution of concepts can be modeled with the logit-linear function, as in Assumption 1? Are such conditional distributions universally expressive, or does the assumption impose constraints on the distribution that can be learned?
2. Does Assumption 2 require that every pair must be differentiated by some $z^r$? Does it require that each $z^r$ must differentiate exactly one pair of concepts?

**Ethical Concerns:**

["NO or VERY MINOR ethics concerns only"]

**Final Justification:**

The authors satisfactorily addressed my concerns.

**Limitations:**

Yes.

**Paper Formatting Concerns:**

None.

**Quality:**

3

**Strengths And Weaknesses:**

Originality: The work introduces a novel but simple latent factor model for describing categories of behaviors in LLMs, under certain modeling assumptions.

Quality & significance: The theory appears to be sound, and the work improves significantly upon the baselines in disentanglement and completeness, another measure of disentanglement. However, it would be desirable to see more evaluation on the validity of the underlying assumptions.

Clarity: the work is generally clearly written.

---

> ### Author Rebuttal · Authors · 2025-07-31
>
> Thank you for your positive feedback on our work. We appreciate that you found our work to be novel, with sound theory and experiments. Below, we answer the questions posed in your review.
>
> ---
>
> #### Expressiveness of log-linear functions
> Good question. Hyvarinen and Morioka (2016) point out that log-linear prediction on top of flexible, highly non-linear feature extraction still has universal expressiveness. Indeed, this style of modeling is akin to nonlinear prediction via kernel regression or basis expansion, which remain flexible, nonlinear predictors.
>
> ---
>
> #### Does Assumption 2 require that every pair must be differentiated by some $z^r$?
> Even though this is not explicitly stated in the assumption, it is implicit in the problem definition that a difference in behavior would correspond to some mediating concept(s) $z^r$.
>
> ---
>
>
> #### Does it require that each $z^r$ must differentiate exactly one pair of concepts?
> No, it is not necessary that each $z^r$ must differentiate exactly one pair of groups. In the refusal case study, for instance, latent dimension 1 (interpreted as harmfulness) is a causal differentiating concept between harmful ($p(refusal|x)=high$) and pseudo-harmful queries ($p(refusal|x)=high$), and also between harmful and harmless queries ($p(refusal|x)=low$).
>
>
> ---
>
> #### References
> [1] Aapo Hyvärinen, Hiroshi Morioka. Unsupervised Feature Extraction by Time-Contrastive Learning and Nonlinear ICA. NeurIPS 2016.

---

> > ### Comment · Reviewer_2QmS · 2025-08-08
> > **Thank you for your response**
> >
> > I am leaving my score unchanged.

---

### Comment · Area_Chair_Nc3j · 2025-08-01

Dear Reviewers,

The author-reviewer discussion phase has started. If you want to discuss with the authors about more concerns and questions, please post your thoughts by adding official comments as soon as possible.

Thanks for your efforts and contributions to NeurIPS 2025.

Best regards,

Your Area Chair

---

### Note · Authors · 2025-08-13

We are grateful to the reviewers for their constructive feedback and discussion. There was a broad agreement on **the soundness of our theoretical contributions** (_“the paper is very mathematically clean”_, _“the theory appears to be sound”_), **the utility of our method** (_“[the paper] offers a new approach to a real thorny issue in causal abstraction”_; _“[the paper] addresses an important challenge in interpretability: extracting concise, causal explanations of LLM behavior without relying on predefined labels or concepts.”_), **and the comprehensiveness of our experiments** (_“the experiments involve several levels of complexity (from clean synthetic data to a more complex collection of real-world data on a relevant topic: refusal behaviour) and compare against well-formed baselines”_).

The main concern that came up in the reviews was regarding the robustness of the assumptions. To alleviate these concerns, we included additional experiments in our rebuttal that demonstrate disentanglement even as we relax the linearity assumptions in LLMs, vary the 1-sparsity assumption, and vary the number of latent variables learned. We hope that these experiments address the questions raised by reviewers, provide practical guidance on deploying our method, and add more confidence in our method.

We would like to finally conclude by saying that the intersection of causal representation learning and language model interpretability is a nascent field. Identifiable unsupervised recovery of factors that dictate LLM behavior is challenging and requires assumptions, as with any causal representation learning method. As we have hopefully shown in our paper and the author-reviewer discussion, our assumptions can practically be met when studying language models and lead to useful insights about their behavior. We believe that our paper can help in making progress towards theoretically grounded language model interpretability.

---

### Decision · Program_Chairs · 2025-09-17

**Decision:**

Accept (spotlight)

**Comment:**

This paper is interesting because it uses causal representation learning to interpret the LLM's behavior. The problem is well-defined, and all details are demonstrated clearly in mathematical format. All reviewers agreed that this paper had a solid theoretical contribution, but also found some issues regarding the assumptions used in the theorem. During the rebuttal, the authors addressed these concerns. It has to be admitted that there will be a gap between theoretical assumptions and real cases. However, the assumptions used in this paper are general in the literature, and the authors also conducted the experiments to further verify their results. Thus, it is worth presenting this paper this time.

All the comments during the rebuttal are excellent, and the authors should merge them into the next version.